# mOSCAR: A Large-scale Multilingual and Multimodal Document-level Corpus

## Abstract

Multimodal Large Language Models (mLLMs) are trained on a large amount of text-image data. While most mLLMs are trained on caption-like data only, Alayrac et al. (2022) showed that additionally training them on interleaved sequences of text and images can lead to the emergence of in-context learning capabilities. However, the dataset they used, M3W, is not public and is only in English. There have been attempts to reproduce their results but the released datasets are English-only. In contrast, current multilingual and multimodal datasets are either composed of caption-like only or medium-scale or fully private data. This limits mLLM research for the 7,000 other languages spoken in the world. We therefore introduce mOSCAR, to the best of our knowledge the first large-scale multilingual and multimodal document corpus crawled from the web. It covers 163 languages, 303M documents, 200B tokens and 1.15B images. We carefully conduct a set of filtering and evaluation steps to make sure mOSCAR is sufficiently safe, diverse and of good quality. We additionally train two types of multilingual model to prove the benefits of mOSCAR: (1) a model trained on a subset of mOSCAR and captioning data and (2) a model trained on captioning data only. The model additionally trained on mOSCAR shows a strong boost in few-shot learning performance across various multilingual image-text tasks and benchmarks, confirming previous findings for English-only mLLMs. The dataset will be made publicly accessible.

## 1 Introduction

Multimodal large language models (mLLMs) are trained on large amounts of text-image data (Radford et al., 2021; Yu et al., 2022; Li et al., 2023; Wang et al., 2023; OpenAI, 2023; Gemini Team et al., 2023; Chameleon Team, 2024). The main paradigm until recently was to train a model from a large collection of web-crawled images and their captions (Li et al., 2021; Wang et al., 2022; Chen et al., 2023b). Models such as Flamingo (Alayrac et al., 2022) challenged this paradigm by being additionally trained on interleaved sequences of text and images from web documents, showing state-of-the-art results on various tasks and in-context learning capabilities that are not present in models trained on caption-like data only. Additionally, McKinzie et al. (2024) recently proved that including interleaved text-image data during training was necessary to get good few-shot learning performance. However, the datasets used to train mLLMs are either private (Alayrac et al., 2022), monolingual or multilingual but only medium-scale (Srinivasan et al., 2021). Some attempts have been made to reproduce these datasets (Zhu et al., 2023; Laurençon et al., 2023) but the resulting datasets are only available in English.

Few image-text datasets are multilingual and most of them are obtained by translating English caption-like datasets, such as multilingual Conceptual Captions (Sharma et al., 2018), into multiple languages using neural machine translation (NMT) systems (Surís et al., 2022; Maaz et al., 2024). This presents some drawbacks such as some languages still being poorly translated by current state-of-the-art NMT models (Liu et al., 2020; Costa-jussà et al., 2022) and some cultural subtleties inherent in each language not being fully conveyed. Some efforts have been conducted to collect large-scale multilingual image captioning datasets, such as LAION-5B (Schuhmann et al., 2022), but they are limited to caption data too, are relatively noisy and more importantly contain a non-negligible share of "not safe for work" (NSFW) content such as pædopornographic images (Schuhmann et al., 2022).

```
Le plus charmant petit square de Paris.

Le square René Viviani a été ouvert au public en 1928. Il occupait l'emplacement de l'annexe de l'Hôtel-Dieu
(il est situé en face de Notre Dame) et d'une rangée de maisons longeant la rue de la Bûcherie. Le conseiller
municipal Léon Riotor et la Commission du Vieux Paris ont particulièrement oeuvré à sa création dans le but de
préserver à la fois le dégagement de l'église Saint-Julien-le-Pauvre et la perspective de Notre-Dame. Quelques
éléments architecturaux parsèment le parc, et contribuent à son charme : pinacles, chapiteaux, balustrades,
provenant sans doute de la cathédrale Notre-Dame. [...]
```

```
Pinacles et chapiteaux ornent le parc : probablement des vestiges de Notre-Dame. La fine poussière derrière la
balustrade de pierre provient du fort vent qui faisait s'envoler le sable et la terre. [...]
```

Figure 1: Example of a French document from mOSCAR.

This motivated us to collect and release the first large-scale multilingual and multimodal document dataset derived from Common Crawl.[1] Our dataset, multimodal OSCAR (mOSCAR), follows the OSCAR initiative (Ortiz Suárez et al., 2019; Abadji et al., 2021; 2022) and covers 303M documents in 163 languages, 200B tokens and 1.15B images. Figure 1 shows an example of a document, more can be found in the Supplementary Material. We carry out extensive filtering to increase its safety and quality. To prove mOSCAR's utility, we train a multilingual OpenFlamingo (Awadalla et al., 2023) from a Gemma-2B language model (Gemma Team et al., 2024) on a subset of mOSCAR and captioning data from LAION-400M (Schuhmann et al., 2021), recaptioned with BLIP (Li et al., 2022), filtered with CLIP (Radford et al., 2021) and translated with NLLB (Costa-jussà et al., 2022). We compare against a similar model trained on captioning data only and show we obtain a strong boost in few-shot learning, confirming previous findings for English (Alayrac et al., 2022; McKinzie et al., 2024; Laurençon et al., 2024). The dataset and models will be made publicly available.

## 2 RELATED WORK

**Large-scale web-based datasets** Numerous datasets have been created by filtering web-crawled data. These include large-scale text-only datasets (Ortiz Suárez et al., 2019; Raffel et al., 2020; Wenzek et al., 2020; Gao et al., 2020; Abadji et al., 2021; Xue et al., 2021; Laurençon et al., 2022; Abadji et al., 2022; Penedo et al., 2023) and multimodal ones (Sharma et al., 2018; Changpinyo et al., 2021; Jia et al., 2021; Schuhmann et al., 2021; 2022; Byeon et al., 2022; Laurençon et al., 2023; Zhu et al., 2023; Gadre et al., 2024). Even if these datasets are not as high quality as smaller and/or hand-crafted ones, they are now the standard to pretrain foundation models, as it has been shown that training bigger models on more data leads to better downstream performances (Brown et al., 2020; Hoffmann et al., 2022; Touvron et al., 2023a;b).

**English image-text datasets** The first open-source image-text datasets were manually created, small-scale and English-only (Ordonez et al., 2011; Lin et al., 2014; Plummer et al., 2015; Krishna et al., 2017). Scaling up these datasets was an appealing solution to overcome limitations of previous image-text models; a few works (Sharma et al., 2018; Changpinyo et al., 2021) proposed to collect millions of image-text pairs from the web before filtering them with well-designed steps. Relaxing the filtering steps enabled the collection of more data and led to large-scale datasets to train image-text foundation models (Radford et al., 2021; Li et al., 2021; Schuhmann et al., 2021; 2022; Byeon et al., 2022). However, these datasets generally contain caption-like image-text pairs only, and it is therefore difficult to observe in-context learning abilities similarly to text-only language models trained on raw documents (Raffel et al., 2020). Alayrac et al. (2022) overcome this issue by training their model directly on documents with interleaved image-text data. While their results are promising, their M3W dataset is English-only and private. Recently, open-source efforts (Zhu et al., 2023; Laurençon et al., 2023) have been made to release a similar dataset but they are still monolingual.

---

[1] https://commoncrawl.org/. The Common Crawl Foundation is a non-profit organization that crawls the web on a monthly basis.

**Multilingual image-text datasets** Only a few image-text datasets are available in multiple languages. One of the first focused on collecting Google images from short queries based on word frequencies from Wikipedia pages in 98 languages (Hewitt et al., 2018). Later, Srinivasan et al. (2021) proposed the WIT dataset, an image-text dataset composed of Wikipedia pages. Although of high quality, it is only medium-scale even for high-resource languages and there are fewer than 50k unique images for most languages. Another approach lies in bootstrapping multilingual and multimodal data from a model trained with English-only data (Mohammed et al., 2023). While effective for captioning, it is computationally expensive to implement in practice. Other multilingual image-text datasets exist but focus on captions only and are highly domain-specific (Kosar et al., 2022; Leong et al., 2022).

## 3 DATASET CREATION PIPELINE

### 3.1 DATA COLLECTION

We collect mOSCAR from the Web ARchive Content (WARC) files of three 2023 Common Crawl dumps, processing them using the FastWARC library (Bevendorff et al., 2021). We remove documents smaller than 500 bytes (50% of the documents), as we find they are usually too small to be considered documents and tend to contain noisy text. We then navigate through the entire Document Object Model (DOM) tree with a depth first search algorithm and ChatNoir library (Bevendorff et al., 2018) to extract nodes of interests corresponding to specific HTML tags.

Following previous work, we extract text from the tags that usually contain the main content of web pages (we refer to them as DOM text nodes), i.e. `<p>`, `<h*>`, `<title>`, `<description>`, `<ul>`, `<ol>`, `<aside>`, `<dl>`, `<dd>`, `<dt>`. Similarly to (Laurençon et al., 2023), we choose to remove `<table>` content as most often it is irrelevant and difficult to render. We extract all `` tags (we refer to them as DOM image nodes). We then remove documents with fewer than 3 text nodes (as they do not contain enough text) and more than 30 image nodes (as we found them to be too noisy).

### 3.2 LANGUAGE IDENTIFICATION

We identify the language of each document using the state-of-the-art open-LID language detector (Burchell et al., 2023), covering 201 languages. We apply open-LID to each DOM text node and keep the three most probable languages with their respective probabilities. The language of the document is then determined by summing over the probabilities of each language detected for each text segment, weighted by the number of characters in the segment[2] and taking the language with the highest score.

### 3.3 TEXT-ONLY FILTERING

We apply a series of filtering steps to the text content of each document independently of the images, with the aim of discarding poor quality documents and cleaning text as best as possible. We first filter at the text-node level and then at the whole document level, before running near-deduplication to keep unique text nodes within a document and unique documents in the dataset.

**Text node filtering** We use a set of heuristics (see Supplementary Material) to extract as much human-generated content as possible while discarding noisy text related to ads and website functions (e.g. "Instagram", "Facebook"). We then keep DOM text nodes with content over 10 bytes. This step, designed to improve the quality of extracted text, removes on average 55% of text nodes.

**Document filtering** We mostly filter "not safe for work" (NSFW) content at the document level. We use an English regular expression to detect adult content, similar to the one used by the Université Toulouse 1 Capitole[3] and remove the entire document if there is a match with any of the DOM text nodes' contents, removing on average 0.5% of documents (mostly English ones). We acknowledge that there is a high probability that this also discards safe content, e.g. we could remove content from

---

[2]This is to avoid mis-assigning the language due to the presence of many short, non-informative DOM text nodes in the same language (e.g. "Cookies", "Subscribe", "Newsletter" etc.) and because language identification is generally less reliable for short segments.

[3]https://dsi.ut-capitole.fr/blacklists/index_en.php

certain communities who use some explicit words in a non-sexual way (Sap et al., 2019). However, we explicitly favour recall over precision to minimise the risk of unsafe content. We additionally remove documents containing fewer than five DOM text nodes and fewer than 300 characters after the previous filtering steps, removing 70.6% of documents.

**Deduplication**   We conduct several types of per-language deduplication at different levels, as this has been shown to improve training efficiency (Abbas et al., 2023). First, we keep unique documents only by removing exact duplicates at the document level. We also remove exact duplicates of text nodes within the same document (4% of text nodes) and near-duplicate text nodes (1% of text nodes) by computing the Levenshtein ratio (Levenshtein, 1966) between all text nodes within the same document and applying a threshold of 0.95. If near-duplicates are found, we keep the first one in the document. Finally, we conduct per language near-deduplication at the document level with MinHashLSH (Broder, 1997; Gionis et al., 1999) following Smith et al. (2022), removing on average 19% of documents:[4] we turn documents into hashing vectors, compute min hashes from these vectors and perform Locality Sensitive Hashing to remove duplicates[5] (see Supplementary Material for more details).

**Toxicity filtering**   Toxic content targeting individuals or groups of people is widespread on the internet and can therefore be found in large-scale web-crawled datasets like mOSCAR without appropriate filtering steps. To alleviate this issue, we apply the same method used by Costa-jussà et al. (2022) and remove documents from mOSCAR based on the presence of a list of "toxic" words for each language[6]. As some words in the list can also be used in a non-toxic way based on the context (e.g.: 'breast' in English), we tag the document as toxic and remove it from mOSCAR if it contains at least two distinct words in the list. This filtering step removes 0.95% of the documents for very high-resource languages (>5M documents), 2.13% for high-resource languages (<5M, >500K), 0.47% for mid-resource languages (<500K, >50K) and 0.64% for low-resource languages (< 50K). When manually analysing 100 random documents removed by this filtering step in each of the 2 (high-resource) languages we are native speakers of (English and French), we found 58 documents with toxic content.

**Personal Identifiable Information**   Personal Identifiable Information (PII) can be found in large-scale web-crawled datasets, we therefore conducted an additional filtering step to replace all detected PII by place holder strings using regular expressions (see Supplementary Material for the list of regular expressions we used). Concretely, we replaced all detected email addresses, phone numbers, credit card numbers, IP addresses and passport numbers.

## 3.4   IMAGE-ONLY FILTERING

We downloaded images from the URLs in DOM image nodes using a modified version of the img2dataset toolkit (Beaumont, 2021) that includes an antivirus scan and follows `robots.txt` instructions to respect the Robots Exclusion Protocol. We then apply a series of filtering steps, first removing images based on heuristics, and then applying multiple NSFW detection models to remove undesirable content. Finally, we conduct a set of deduplication steps.

**Rule-based filters**   Similarly to previous works (Schuhmann et al., 2021) and to avoid extracting low-resolution images and favicons, we keep images with a minimum height and width of 150 pixels. We restrict the aspect ratio to be between 3 and 1/3 (to remove banners), we remove images if their URLs contain the words "logo", "banner", "button", "widget", "icon" or "plugin" or if the image name from the URL matches "twitter", "facebook" or "rss" (to remove logos). This step removes 13.6% of the URLs. At this stage, we downloaded 2.5B images with an average success rate of 55%.

**NSFW detection**   We use multiple NSFW automatic models to remove as much unsafe content as possible. We first combine two NSFW detectors: nsfw-detector (Laborde), a 5-class classifier with a

---

[4]With some disparity among languages as we found more duplicates for low- than high-resource languages.

[5]We performed this using the `datasketch` python library.

[6]The list of these words for each language can be found here: `https://github.com/facebookresearch/flores/tree/main/toxicity`

MobileNet (Howard et al., 2017) backbone fine-tuned on 60GB of annotated data and NudeNet,[7] an object detector trained to detect different types of nudity in images. We combined the two models as we found the first to be gender-biased while the second gives a large number of false positives for non-human images. Concretely, we consider an image an NSFW candidate if the sum of the probabilities for the classes 'porn' and 'hentai' is superior to 0.8 using nsfw-detector. We then tag the image as NSFW if one of the sensitive 'exposed' classes of NudeNet gets a probability superior to 0.5. If a document contains an image with an NSFW tag, we remove the entire document from the dataset, which removes 0.5% of images. We manually inspecting 1,000 images of the remaining data and found no NSFW content. We manually inspected 1,000 images of the removed content and found 63.4% of NSFW images.

**CSAM content** Child Sexual Abuse Material (CSAM) is widespread on the internet and is therefore likely to be found in such a large-scale dataset crawled from the web. Removing CSAM is challenging as there is no training data nor open-source detection models available as these could be used in a harmful way. We rely on Thorn's CSAM classifier[8], a proprietary classifier trained to detect CSAM content in images. We tag the image as CSAM if the probability of the class CSAM is superior to 0.4 to favour recall over precision. As mentioned above, if a document contains an image with a CSAM tag, we remove it from the dataset. This step removes 0.07% of the images.

**Deduplication** To avoid memorisation issues often seen in models trained on datasets with many duplicated images (Somepalli et al., 2023; Carlini et al., 2023; Webster et al., 2023; Somepalli et al., 2024), we perform deduplication at the image level. We first remove duplicate images within the same document by URL matching (removing 8.7% of URLs). We then compute a perceptual hash (pHash) for each image using the imagehash library[9] and remove images with the same pHash within the same document, keeping only the first occurrence. We also limit the number of times an image can appear in the dataset per-language to 10 using both URL matching and perceptual hashing (this removes 2.5% of images). We do this per-language and not across languages as having the same images in documents from different languages could encourage cross-lingual transfer.

**Personal Identification Information** To protect PII in images, we use a lightweight face detector[10] and apply a threshold of 0.99 to detect faces in the images. We apply such a high threshold as we found the model to be biased towards detecting faces with high probability in images without any human. For each image in mOSCAR, we distribute the bounding boxes of the detected faces so that users can blur them when downloading the images. More details are provided in the Supplementary Material.

## 3.5 DATA DECONTAMINATION

LLMs and mLLMs are trained on web-crawled data that can contain the benchmarks they are tested on (Dodge et al., 2021). As they are good at memorizing training data (Carlini et al., 2023), this data contamination is problematic. We therefore discard all images with the same perceptual hash as any of the images from the evaluation benchmarks (and their training sets) we use (see Section 5.1). This step removes on average 126,016 images for high-resource languages (up to 300K images for English), 6,862 images for mid-resource languages and 45 images for low-resource languages.

## 3.6 TEXT-IMAGE JOINT FILTERING

Our aim is to obtain truly multimodal documents where all images are related to at least one of the text nodes in some way[11] and vice versa. We choose to apply joint text-image filtering to discard images and/or text nodes that are irrelevant to the rest of the document (e.g. the case of ads and website functionalities). To do this, we use NLLB-SIGLIP[12] (Visheratin, 2023), a multilingual version

---

[7]https://github.com/vladmandic/nudenet
[8]https://safer.io/
[9]https://github.com/JohannesBuchner/imagehash
[10]https://github.com/Linzaer/Ultra-Light-Fast-Generic-Face-Detector-1MB
[11]We do not limit ourselves to caption-like relation and instead allow all types of text-image relation.
[12]siglip-base-patch16-224 as vision encoder and nllb-distilled-600M as text encoder.

of SIGLIP (Zhai et al., 2023) trained with the encoder of NLLB (Costa-jussà et al., 2022), which covers all mOSCAR languages.[13] We compute cosine similarity scores between all images and all paragraphs[14] within a same document. To remove irrelevant text nodes or images in a document, we mimic a text-image retrieval task, which means we avoid using arbitrary cosine similarity thresholds for each language and can reduce length biases and those in favour of caption-like paragraphs. For each candidate pair we randomly sample 63 negative images and 63 negative similar-length paragraphs from the same language but other documents. We tag the text node (resp. image) as valid if the cosine similarity of the pair is among the top 8 of the text-to-image (resp. image-to-text) similarity scores computed with the candidate text node (resp. image) and all the negative images (resp. text nodes). This means that we tag the text node (resp. image) as valid if it has a significantly higher score than a score computed with a random image (resp. text) for at least one of the images (resp. text node) in the document. We then discard text nodes and images not tagged as valid (on average 35% of the DOM text nodes and 10% of the images within a document).

After this filtering step, we apply additional text-only filters to keep documents superior to 100 bytes. We also reapply the open-lid language detector (Burchell et al., 2023) as described in Section 3.2 as we found the last filtering step to change the major language of some documents.

# 4 MULTIMODAL OPEN SUPER-LARGE CRAWLED AGGREGATED CORPUS (MOSCAR)

mOSCAR is extracted from three Common Crawl dumps from 2023. Due to computational constraints and in order to extract a maximum number of documents for low-resource languages, we extracted all languages from the first dump only. We removed the 6 most high-resource languages from the second dump and only extracted the languages with fewer than 1M documents for the last dump. Table 1 shows a distribution of the total number of languages and their number of documents.

To avoid data poisoning (Carlini et al., 2024), we release a hash (sha512) with each mOSCAR image.

| #documents | 10M | 5M | 1M | 500K | 200K | 50K | 10K | 5K | 1K |
|---|---|---|---|---|---|---|---|---|---|
| #languages | 10 | 15 | 36 | 46 | 56 | 75 | 119 | 133 | 154 |

Table 1: Number of languages with at least $N$ documents

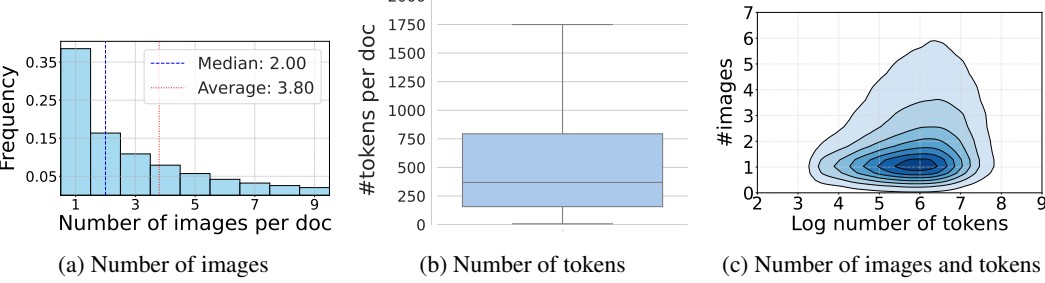

| (a) Number of images | (b) Number of tokens | (c) Number of images and tokens |
|---|---|---|

Figure 2: Distributions of numbers of tokens and images per document

mOSCAR is composed of 303M documents (200B tokens, 1.15B images) from 163 languages. Figure 2 shows the distribution of images and tokens per document and their joint distribution. As shown in Figure 2a, the mean and median number of images per document is 2 and 3.80.

## 4.1 QUALITY VS DIVERSITY

While improving overall data quality, the filtering steps we applied (see Section 3) necessarily have a negative impact on diversity. We therefore study the trade-off between quality and diversity and compare against previously published, well-used datasets.

---

[13]We use the open-clip (Ilharco et al., 2021) model version and the transformers (Wolf et al., 2020) library.

[14]We refer to paragraph as the text content in a DOM text node.

### 4.1.1 TEXT CONTENT

**Diversity**   By contruction, mOSCAR is diverse in terms of number of languages, so we focus on the diversity of mOSCAR's English documents and compare against mmc4 (Zhu et al., 2023), OBELICS (Laurençon et al., 2023) and the English subset of WIT (Srinivasan et al., 2021). We compute the Vendi score (Friedman and Dieng, 2023) on a set of SimCSE embeddings (Gao et al., 2021) with a RoBERTa encoder (Liu et al., 2019) to evaluate the content diversity. Since embedding-based diversity metrics target content diversity well but are less relevant for lexical diversity (Tevet and Berant, 2021), we measure lexical diversity via the distinct $n$-gram ratio (Li et al., 2016).

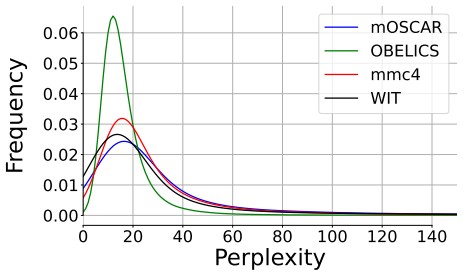

Figure 3: Perplexity of 100K random documents from different datasets.

|  | Vendi score | Dist. $n$-gram ratio |
|---|---|---|
| mOSCAR | 69.05 ($\pm$ 0.14) | 0.472 ($\pm$ 0.002) |
| mmc4 | 67.93 ($\pm$ 0.12) | 0.494 ($\pm$ 0.002) |
| OBELICS | 58.49 ($\pm$ 0.09) | 0.488 ($\pm$ 0.001) |
| WIT | **73.30** ($\pm$ 0.09) | **0.530** ($\pm$ 0.001) |

Table 2: Average text diversity scores ($\pm$ standard error) of text documents.

**Comparison with other datasets**   For content diversity, we randomly sample 30M documents for mOSCAR, mmc4 and OBELICS and 3M documents for WIT and represent the documents by their SimCSE embedding. We compute the Vendi Score with cosine similarity on a randomly sampled subset of 65,536 documents. Table 2 shows that mOSCAR English content is more diverse than mmc4 and OBELICS but less diverse than WIT. For lexical diversity, we randomly sample 3M documents for mOSCAR, mmc4, OBELICS and WIT and compute the distinct $n$-gram ratio on a subset of 8,192 documents for $n$ from 1 to 4. Table 2 shows that mOSCAR is slightly less lexically diverse than OBELICS and mmc4, while WIT is by far the most diverse.

**Quality**   To evaluate document quality, we focus on English documents and compute their perplexity using Gemma-2B (Gemma Team et al., 2024). Figure 3 shows the kernel density estimation of the distribution of the perplexity of 100K randomly sampled documents from different datasets: mOSCAR is comparable to mmc4 and WIT, while OBELICS appears to be the of the highest quality. mOSCAR is therefore comparable to other interleaved image-text dataset in terms of quality and diversity of its English subset. It is however more diverse than English-only datasets by its multilingual construction and more than 10 times larger than existing multilingual interleaved image-text datasets such as WIT.

### 4.1.2 IMAGE DIVERSITY

| mOSCAR | LAION-400M | WIT | | English | All |
|---|---|---|---|---|---|
| 55.74 ($\pm$ 0.16) | **67.59** ($\pm$ 0.16) | 36.14 ($\pm$ 0.08) | | 52.36 ($\pm$ 0.18) | **54.78** ($\pm$ 2.29) |
| (a) Comparison of different datasets. | | | | (b) mOSCAR (English vs. any language). | |

Table 3: Average Vendi score ($\pm$ standard error) of images.

**Comparison with other datasets**   We compute the Vendi Score on random samples of images for different datasets, comparing the images from English mOSCAR documents with those from Conceptual Captions (Changpinyo et al., 2021), LAION-400M (Schuhmann et al., 2021) and WIT (Srinivasan et al., 2021). We represent each image by its SigLIP[15] (Zhai et al., 2023) embedding and compute the Vendi score on batches of size 65,536 and a total of 1M images for each dataset. In

---

[15]We use `siglip-base-patch16-224`.

Table 3a, we notice that the set of images in mOSCAR documents are more diverse than images from WIT documents but less diverse than LAION-400M.

**Multilingual diversity** We also compare the diversity of images from English documents and of images sampled from documents of any language (English included). We use multilingual SigLIP (Chen et al., 2023a) trained on WebLI (Chen et al., 2023b) to compute image embeddings used to get the Vendi score. We again use a batch of size 65,536 and a total of 3M images, and we do not sample multiple images from a same document. For the multilingual setting, we randomly sample 50 languages and an equal number of images for each language to build the batch. As we did not do any image deduplication across languages, we could expect to have less diversity in the multilingual setting. However, Table 3b shows that the set of images is on average more diverse when sampled from all documents than from English-only documents. This means that the distribution of images is not exactly the same across languages, potentially due to cultural differences.

## 5 TRAINING A MULTILINGUAL MULTIMODAL LANGUAGE MODEL

We train a multilingual Flamingo-like model on mOSCAR that we call multilingual Open Flamingo. As adding captioning data to training data has been shown to improve zero-shot performance (McKinzie et al., 2024), we additionally train on LAION-400M, which we re-captioned using BLIP (Li et al., 2022), filtered with CLIP score (Radford et al., 2021) and translated using distilled NLLB-600M (Visheratin, 2023) following the proportion of languages found in mOSCAR. We use Gemma-2B (Gemma Team et al., 2024) as the underlying language model and we train the model on 50M mOSCAR documents and 100M randomly sampled image-text pairs. We also train a model on 300M image-text pairs, a model trained on 35M WIT (Srinivasan et al., 2021) documents and 70M text-image pairs and a model trained on 50M mOSCAR documents from the English subset and 100M English image-text pairs as comparison baselines. We additionally compare with OpenFlamingo-3B-MPT (Awadalla et al., 2023) as the *translate-test* baseline. The full list of languages for training and the implementation details can be found in the Supplementary Material.

### 5.1 EVALUATION SETUP

We evaluate the models using a broad set of image-text multilingual tasks and benchmarks. We use the IGLUE benchmark (Bugliarello et al., 2022) composed of XVNLI, MaRVL (Liu et al., 2021) to test reasoning, xGQA (Pfeiffer et al., 2022) to test visual question answering capabilities and xFlickr&CO (Young et al., 2014; Karpathy and Fei-Fei, 2015; Yoshikawa et al., 2017) for captioning. We also include Crossmodal-3600 (XM3600) (Thapliyal et al., 2022) and MaXM (Changpinyo et al., 2022) as they cover a broader range of languages. To test to what extent models trained on mOSCAR can perform zero-shot multimodal machine translation (MMT), we also test on Multi30K (Elliott et al., 2016; 2017; Barrault et al., 2018) and CoMMuTE (Futeral et al., 2023). For captioning we compute the CideR (Vedantam et al., 2015) score and we tokenize references and model outputs with the Stanford Core NLP tokenizer for English and Stanza (Qi et al., 2020) tokenizers for other languages. To evaluate Multi30k, we compute BLEU (Papineni et al., 2002) score from Sacrebleu (Post, 2018) with *13a* tokenization and default parameters. We use accuracy for CoMMuTE. More details can be found in the Supplementary Material.

### 5.2 RESULTS

Tables 4 and 6 show the average results across all languages. Full results are available in the Supplementary Material. We notice that the multilingual OpenFlamingo trained additionally on mOSCAR gets better results than the model trained on captioning data only while having seen fewer image-text pairs during training. More importantly, when increasing the number of few-shot examples from 0 to 16, it sees gains of on average +6.71 points on VQA benchmarks and +19.39 CideR points on captioning benchmarks. In contrast, the model trained on text-image pairs only sees gains of +2.82 and +9.08 points respectively. In cross-modal machine translation, the model additionally trained on interleaved data is again much better than the one trained on just captioning data, which is not able to translate the Multi30k benchmark at all.[16] Moreover, mOSCAR helps the model to learn to zero-shot

---

[16]Most of the time, the model is not able to follow the prompt and only outputs the end of sequence token.

disambiguate translations as shown by the improved average score on CoMMuTE (63.54) compared to the model trained on captions only (61.36).

Multilingual Open Flamingo trained on mOSCAR & text-image pairs is also better than Open-Flamingo 3B MPT evaluated on translate test benchmarks[17]. However, we obtain the best results (except for MaXM) by evaluating our multilingual Open Flamingo on the translate-test benchmarks since the underlying language model (Gemma-2B) is far better in English than other languages. We also notice that all models struggle with reasoning classification tasks (MaRVL, XVNLI) where they obtain scores close to random guessing.

Table 5 additionally shows that Multilingual Open Flamingo trained on mOSCAR obtains much better results than the same model trained on WIT for equivalent training data seen during training[18] (except for Multi30K benchmark) which means mOSCAR is better suited than WIT for training multilingual mLLMs. Eventually, Table 7 shows that we don't face a drop in performances in English performances when training the model on 43 languages (multilingual Open Flamingo) in comparison to training it on the English subset of mOSCAR and English text-image pairs.

Additional comparison results with InternVL2 (Chen et al., 2024), Llava-NeXT (Li et al., 2024), PaliGemma (Beyer* et al., 2024) and Idefics2 (Laurençon et al., 2024) can be found in the Supplementary Material.

| | #shots | xFlickR&CO | XM3600 | xGQA | MaXM | MaRVL | XVNLI | Multi30K | CoMMuTE |
|---|---|---|---|---|---|---|---|---|---|
| Multi. OF *mOSCAR + cap.* | 0 | 16.91 | 7.45 | 26.95 | 22.33 | 49.56 | 33.88 | 22.91 | 63.34 |
| | 4 | 34.80 | 22.18 | 32.33 | 26.33 | 49.64 | 34.07 | 23.27 | 63.22 |
| | 8 | 36.90 | 23.48 | 34.24 | 27.08 | **51.48** | **36.60** | 23.59 | **63.54** |
| | 16 | **39.46** | **23.67** | **35.23** | **27.47** | 49.84 | 34.85 | **23.85** | 62.78 |
| Multi. OF *cap. only* | 0 | 9.57 | 4.21 | 8.62 | 4.01 | **49.88** | 33.76 | 0.00 | 61.36 |
| | 4 | 13.20 | 9.26 | 13.45 | 4.15 | 49.54 | 32.04 | 0.00 | 61.13 |
| | 8 | 18.00 | 10.35 | 12.82 | 4.88 | 49.65 | 33.71 | 0.01 | 60.90 |
| | 16 | 19.87 | 12.07 | 13.37 | 4.89 | 49.79 | 32.70 | 0.74 | 60.25 |

Table 4: Results averaged over all languages. Multi. OF refers to multilingual Open Flamingo, *mOSCAR + cap.* refers to the model trained on text-image pairs and mOSCAR while *cap. only* refers to the model trained only on text-image pairs. **Bold** is best result.

| | #shots | xFlickR&CO | XM3600 | xGQA | MaXM | MaRVL | XVNLI | Multi30K | CoMMuTE |
|---|---|---|---|---|---|---|---|---|---|
| Multi. OF (35M) *mOSCAR + cap.* | 0 | 19.07 | 8.73 | 25.08 | 19.64 | **49.77** | 33.01 | 22.70 | **63.75** |
| | 4 | 34.32 | 20.59 | 31.90 | 23.90 | 49.67 | 36.07 | 22.79 | 63.65 |
| | 8 | 36.77 | 22.15 | 33.9 | 24.41 | 49.72 | **37.16** | 23.21 | 63.00 |
| | 16 | **37.63** | **22.24** | **35.71** | **25.38** | 49.73 | 35.36 | 23.48 | 62.77 |
| Multi. OF (35M) *WIT + cap.* | 0 | 9.39 | 4.67 | 19.81 | 14.63 | 49.71 | 32.78 | **26.99** | 56.75 |
| | 4 | 7.68 | 2.99 | 25.68 | 16.12 | 49.72 | 33.51 | **26.99** | 53.27 |
| | 8 | 8.91 | 3.63 | 27.06 | 16.81 | 49.74 | 32.77 | **26.99** | 55.33 |
| | 16 | 9.74 | 4.14 | 28.14 | 16.34 | 49.74 | 33.63 | **26.99** | 54.04 |

Table 5: Results averaged over all languages and comparison between a model trained on WIT and a checkpoint of multilingual Open Flamingo trained on 35M mOSCAR documents (full model was trained on 50M mOSCAR documents). Both models were trained on 35M documents from their respective training datasets and 70M text-image pairs for fair comparison. Multi. OF (35M) refers to multilingual Open Flamingo trained on 35M documents. **Bold** is best result.

We additionally compare results at different training steps, defined by the number of images seen during training. Figure 4 shows the difference of averaged scores between the model trained on all data and the model trained only on text-images pairs. We notice that the gap first decreases until 20M images seen and keep increasing over training at all training steps after that. Particularly, the gap is wider for few-shot learning.

---

[17]This means benchmarks were translated from local languages to English, using Google Translate API

[18]We select the checkpoint of multilingual Open Flamingo trained on 35M documents and 70M captions to have fair comparison.

|  | #shots | xGQA | MaXM | MaRVL | XVNLI |
|---|---|---|---|---|---|
| OF-3B MPT | 0 | 18.34 | 7.68 | 49.75 | 32.73 |
|  | 4 | 22.97 | 7.82 | 49.70 | 35.82 |
|  | 8 | 28.57 | 8.32 | 49.71 | 31.29 |
|  | 16 | 31.82 | 9.04 | 49.72 | 33.29 |
| Multi. OF *mOSCAR + cap.* | 0 | 30.16 | 10.06 | 49.93 | 34.66 |
|  | 4 | 35.55 | 9.89 | 48.99 | 36.10 |
|  | 8 | 36.78 | 10.12 | **50.54** | **39.69** |
|  | 16 | **37.75** | **11.49** | 49.57 | 37.97 |

Table 6: *Translate-test* results averaged over languages where all benchmarks were translated from local languages into English using Google Translate API. Multi. OF *mOSCAR + cap.* refers to Multilingual Open Flamingo trained on mOSCAR and text-image pairs while OF-3B MPT refers to Open Flamingo (Awadalla et al., 2023) based on MPT (Team, 2023) and trained on mmc4 (Zhu et al., 2023) and English text-image pairs.

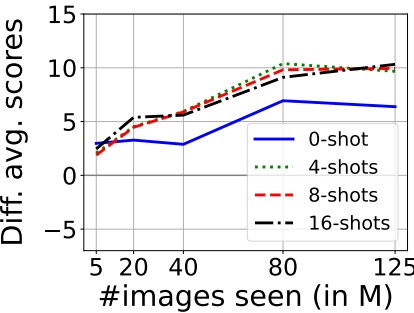

Figure 4: Score differences averaged over benchmarks and languages between the model trained on mOSCAR + text-image pairs and the model trained only on text-image pairs. **Bold** is best result.

|  | #shots | xFlickR&CO | XM3600 | xGQA | MaXM | XVNLI |
|---|---|---|---|---|---|---|
| Multilingual OF *mOSCAR + cap.* | 0 | 29.64 | 42.57 | 34.24 | 36.58 | 34.62 |
|  | 4 | 51.47 | 77.98 | 37.91 | 38.13 | 33.59 |
|  | 8 | 56.75 | 77.64 | 39.44 | **38.52** | **38.75** |
|  | 16 | **59.89** | 78.18 | 40.09 | 35.80 | 36.60 |
| English OF *English mOSCAR + English cap.* | 0 | 32.70 | 43.75 | 34.71 | 36.19 | 35.82 |
|  | 4 | 51.39 | 75.33 | 37.48 | 37.96 | 34.88 |
|  | 8 | 51.44 | 77.73 | 39.64 | 38.35 | 36.86 |
|  | 16 | 59.24 | **78.38** | **40.36** | 37.35 | 37.11 |

Table 7: Results on the English subsets of the test sets and comparison between multilingual Open Flamingo and an Open Flamingo trained on the English subset of mOSCAR and English text-image pairs (English OF). Both models were trained on 50M documents from their respective training datasets and 100M text-image pairs for fair comparison. **Bold** is best result.

## 6 CONCLUSION, LIMITATIONS AND SOCIETAL IMPACTS

We introduce mOSCAR, a large-scale multilingual and multimodal dataset covering 163 languages and composed of 303M documents, 200B tokens and 1.15B images. We show that mOSCAR is of good quality, diverse and can be used to train a multilingual and multimodal LLM. We ensure that mOSCAR is as safe as possible by applying a series of filtering steps to remove NSFW and toxic content. We however did not conduct any analysis of its biases as this is challenging in a multilingual setting. As it is crawled from the internet, it is indeed possible that mOSCAR reflects biases widespread on it. Training a model on mOSCAR must therefore be combined with additional alignment training steps to mitigate potential biases towards groups of people. Nevertheless, by its multilingual nature, mOSCAR is a step towards the inclusion of more languages, cultures and people in accessing mLLMs.

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
