# A  APPENDIX

## A.1  MOSCAR LANGUAGES & STATISTICS

| Languages | | | | Statistics | | |
|---|---|---|---|---|---|---|
| Lang. name | Code | Family | Script | #documents | #images | #tokens |
| Acehnese | ace_Latn | Austronesian | Latin | 2,159 | 9,026 | 1,395,381 |
| Mesopotamian Arabic | acm_Arab | Afro-Asiatic | Arabic | 1,282 | 5,621 | 704,549 |
| Tunisian Arabic | aeb_Arab | Afro-Asiatic | Arabic | 5,933 | 34,270 | 2,308,455 |
| Afrikaans | afr_Latn | Indo-European | Latin | 50,061 | 211,876 | 38,761,504 |
| South Levantine Arabic | ajp_Arab | Afro-Asiatic | Arabic | 8,603 | 69,051 | 3,869,688 |
| Tosk Albanian | als_Latn | Indo-European | Latin | 856,144 | 2,543,758 | 441,244,377 |
| Amharic | amh_Ethi | Afro-Asiatic | Ge'ez | 39,031 | 149,739 | 33,768,732 |
| North Levantine Arabic | apc_Arab | Afro-Asiatic | Arabic | 16,198 | 110,792 | 8,268,237 |
| Modern Standard Arabic | arb_Arab | Afro-Asiatic | Arabic | 3,794,792 | 14,757,353 | 3,346,786,610 |
| Najdi Arabic | ars_Arab | Afro-Asiatic | Arabic | 52,102 | 261,275 | 39,066,487 |
| Moroccan Arabic | ary_Arab | Afro-Asiatic | Arabic | 117,957 | 584,301 | 188,462,338 |
| Egyptian Arabic | arz_Arab | Afro-Asiatic | Arabic | 761,113 | 3,785,164 | 635,018,784 |
| Assamese | asm_Beng | Indo-European | Bengali | 2,947 | 7,228 | 543,676 |
| Asturian | ast_Latn | Indo-European | Latin | 87,649 | 533,723 | 25,499,269 |
| Awadhi | awa_Deva | Indo-European | Devanagari | 8,179 | 29,142 | 2,293,620 |
| Central Aymara | ayr_Latn | Aymaran | Latin | 10,112 | 57,294 | 2,343,403 |
| South Azerbaijani | azb_Arab | Turkic | Arabic | 3,411 | 14,825 | 3,143,946 |
| North Azerbaijani | azj_Latn | Turkic | Latin | 511,832 | 1,796,046 | 256,160,442 |
| Bashkir | bak_Cyrl | Turkic | Cyrillic | 3,287 | 12,031 | 2,600,135 |
| Bambara | bam_Latn | Manding | Latin | 3,011 | 17,666 | 446,961 |
| Balinese | ban_Latn | Austronesian | Latin | 787 | 4,894 | 392,978 |
| Belarusian | bel_Cyrl | Indo-European | Cyrillic | 60,443 | 276,672 | 71,854,171 |
| Bemba | bem_Latn | Atlantic–Congo | Latin | 582 | 3,018 | 1,021,026 |
| Bengali | ben_Beng | Indo-European | Bengali | 204,475 | 758,222 | 30,400,395 |
| Bhojpuri | bho_Deva | Indo-European | Devanagari | 4,190 | 18,339 | 715,786 |
| Banjar | bjn_Latn | Austronesian | Latin | 1,764 | 9,017 | 1,093,443 |
| Bosnian | bos_Latn | Indo-European | Latin | 635,750 | 2,642,491 | 423,073,661 |
| Buginese | bug_Latn | Austronesian | Latin | 584 | 2,379 | 167,459 |
| Bulgarian | bul_Cyrl | Indo-European | Cyrillic | 2,578,191 | 11,601,214 | 1,736,106,287 |
| Catalan | cat_Latn | Indo-European | Latin | 1,132,056 | 4,638,966 | 598,942,711 |
| Cebuano | ceb_Latn | Austronesian | Latin | 14,924 | 75,258 | 10,221,371 |
| Czech | ces_Latn | Indo-European | Latin | 3,736,126 | 12,683,461 | 2,767,295,966 |
| Central Kurdish | ckb_Arab | Indo-European | Arabic | 36,413 | 135,461 | 21,622,335 |
| Crimean Tatar | crh_Latn | Turkic | Latin | 2,744 | 10,079 | 1,173,321 |
| Welsh | cym_Latn | Indo-European | Latin | 38,616 | 155,591 | 27,237,252 |
| Danish | dan_Latn | Indo-European | Latin | 2,020,516 | 9,214,031 | 1,207,829,704 |
| German | deu_Latn | Indo-European | Latin | 20,265,504 | 86,393,702 | 8,315,212,019 |
| Southwestern Dinka | dik_Latn | Nilo-Saharan | Latin | 1,233 | 4,766 | 1,098,795 |
| Greek | ell_Grek | Indo-European | Greek | 4,895,433 | 15,147,284 | 2,909,427,055 |
| English | eng_Latn | Indo-European | Latin | 51,658,029 | 205,363,181 | 32,599,001,993 |
| Esperanto | epo_Latn | Artificial | Latin | 23,619 | 112,577 | 26,976,847 |
| Estonian | est_Latn | Uralic | Latin | 1,022,368 | 5,108,102 | 589,045,973 |
| Basque | eus_Latn | Isolate | Latin | 682,599 | 2,914,120 | 259,930,954 |
| Faroese | fao_Latn | Indo-European | Latin | 14,921 | 56,934 | 6,579,921 |
| Fijian | fij_Latn | Austronesian | Latin | 1,039 | 4,039 | 416,670 |
| Finnish | fin_Latn | Uralic | Latin | 2,377,155 | 10,263,171 | 1,749,904,041 |
| French | fra_Latn | Indo-European | Latin | 19,963,542 | 76,851,982 | 13,818,099,493 |
| Friulian | fur_Latn | Indo-European | Latin | 15,823 | 120,878 | 2,550,209 |
| Nigerian Fulfulde | fuv_Latn | Atlantic-Congo | Latin | 919 | 4,281 | 264,234 |
| West Central Oromo | gaz_Latn | Afro-Asiatic | Latin | 3,399 | 9,071 | 1,640,693 |
| Scottish Gaelic | gla_Latn | Indo-European | Latin | 19,638 | 105,937 | 13,119,348 |
| Irish | gle_Latn | Indo-European | Latin | 60,303 | 267,562 | 45,341,371 |
| Galician | glg_Latn | Indo-European | Latin | 410,489 | 1,696,763 | 197,685,077 |
| Guarani | grn_Latn | Tupian | Latin | 207,800 | 1,038,296 | 48,610,979 |
| Gujarati | guj_Gujr | Indo-European | Gujarati | 21,916 | 87,805 | 3,202,096 |
| Haitian Creole | hat_Latn | Indo-European | Latin | 105,777 | 667,801 | 34,261,838 |

| Languages | | | | Statistics | | |
|---|---|---|---|---|---|---|
| Lang. name | Code | Family | Script | #documents | #images | #tokens |
| Hausa | hau_Latn | Afro-Asiatic | Latin | 21,850 | 81,141 | 11,807,898 |
| Hebrew | heb_Hebr | Afro-Asiatic | Hebrew | 1,098,800 | 4,708,947 | 859,238,720 |
| Hindi | hin_Deva | Indo-European | Devanagari | 543,928 | 1,745,222 | 118,903,998 |
| Chhattisgarhi | hne_Deva | Indo-European | Devanagari | 832 | 3,908 | 205,345 |
| Croatian | hrv_Latn | Indo-European | Latin | 1,689,553 | 8,315,237 | 998,928,993 |
| Hungarian | hun_Latn | Uralic | Latin | 3,515,058 | 15,293,132 | 2,811,446,583 |
| Armenian | hye_Armn | Indo-European | Armenian | 336,285 | 1,126,920 | 199,883,484 |
| Igbo | ibo_Latn | Atlantic-Congo | Latin | 7,089 | 41,672 | 3,014,602 |
| Ilocano | ilo_Latn | Austronesian | Latin | 7,076 | 59,327 | 832,454 |
| Indonesian | ind_Latn | Austronesian | Latin | 6,644,918 | 16,237,247 | 2,895,956,979 |
| Icelandic | isl_Latn | Indo-European | Latin | 239,195 | 1,003,522 | 131,308,802 |
| Italian | ita_Latn | Indo-European | Latin | 12,812,932 | 47,011,085 | 8,144,757,759 |
| Javanese | jav_Latn | Austronesian | Latin | 18,192 | 100,952 | 15,206,708 |
| Japanese | jpn_Jpan | Japonic | Kanji | 14,154,575 | 23,435,549 | 8,539,956,266 |
| Kabyle | kab_Latn | Afro-Asiatic | Latin | 6,101 | 33,923 | 1,781,992 |
| Kannada | kan_Knda | Dravidian | Kannada | 9,373 | 33,147 | 1,206,651 |
| Kashmiri | kas_Arab | Indo-European | Arabic | 1,498 | 5,284 | 3,384,394 |
| Georgian | kat_Geor | Kartvelian | Georgian | 353,471 | 1,300,710 | 274,042,522 |
| Kazakh | kaz_Cyrl | Turkic | Cyrillic | 248,403 | 718,126 | 138,597,176 |
| Halh Mongolian | khk_Cyrl | Mongolic | Cyrillic | 123,789 | 505,098 | 83,628,495 |
| Khmer | khm_Khmr | Austroasiatic | Kher | 23,348 | 116,437 | 2,915,205 |
| Kinyarwanda | kin_Latn | Atlantic-Congo | Latin | 20,381 | 108,280 | 10,268,334 |
| Kyrgyz | kir_Cyrl | Uralic | Cyrillic | 51,221 | 194,092 | 33,981,180 |
| Northern Kurdish | kmr_Latn | Indo-European | Latin | 34,593 | 142,634 | 21,972,155 |
| Korean | kor_Hang | Koreanic | Hanja | 2,614,038 | 13,562,957 | 2,000,344,511 |
| Lao | lao_Laoo | Kra-Dai | Lao | 49,925 | 205,452 | 30,098,274 |
| Ligurian | lij_Latn | Indo-European | Latin | 3,581 | 26,740 | 1,046,463 |
| Limburgish | lim_Latn | Indo-European | Latin | 70,099 | 443,903 | 25,465,590 |
| Lingala | lin_Latn | Atlantic-Congo | Latin | 6,304 | 41,400 | 1,580,536 |
| Lithuanian | lit_Latn | Indo-European | Latin | 1,673,790 | 8,772,570 | 1,153,604,941 |
| Lombard | lmo_Latn | Indo-European | Latin | 14,053 | 61,359 | 6,270,646 |
| Latgalian | ltg_Latn | Indo-European | Latin | 5,174 | 21,062 | 2,903,043 |
| Luxembourgish | ltz_Latn | Indo-European | Latin | 27,946 | 142,470 | 13,925,521 |
| Ganda | lug_Latn | Afro-Asiatic | Latin | 1,475 | 4,118 | 688,308 |
| Mizo | lus_Latn | Sino-Tibetan | Latin | 7,009 | 22,630 | 4,106,536 |
| Standard Latvian | lvs_Latn | Indo-European | Latin | 857,757 | 3,937,940 | 578,441,751 |
| Magahi | mag_Deva | Indo-European | Devanagari | 290 | 1,088 | 94,031 |
| Malayalam | mal_Mlym | Dravidian | Malayalam | 11,203 | 44,417 | 1,420,906 |
| Marathi | mar_Deva | Indo-European | Devanagari | 43,720 | 142,001 | 6,164,176 |
| Minangkabau | min_Latn | Austronesian | Latin | 1,523 | 7,300 | 447,320 |
| Macedonian | mkd_Cyrl | Indo-European | Cyrillic | 539,149 | 1,841,846 | 304,592,615 |
| Maltese | mlt_Latn | Afro-Asiatic | Latin | 56,666 | 327,331 | 27,114,870 |
| Maori | mri_Latn | Austronesian | Latin | 20,840 | 114,680 | 24,524,962 |
| Burmese | mya_Mymr | Sino-Tibetan | Mon | 6,575 | 36,661 | 406,016 |
| Dutch | nld_Latn | Indo-European | Latin | 16,890,074 | 64,609,055 | 9,493,533,101 |
| Norwegian Nynorsk | nno_Latn | Indo-European | Latin | 138,384 | 701,972 | 57,812,652 |
| Norwegian Bokmål | nob_Latn | Indo-European | Latin | 2,192,012 | 9,534,178 | 1,267,421,216 |
| Nepali | npi_Deva | Indo-European | Devanagari | 28,042 | 116,363 | 2,892,865 |
| Nyanja | nya_Latn | Atlantic-Congo | Latin | 11,749 | 65,324 | 8,513,823 |
| Occitan | oci_Latn | Indo-European | Latin | 61,681 | 323,632 | 21,029,975 |
| Odia | ory_Orya | Indo-European | Odia | 3,759 | 14,373 | 340,695 |
| Pangasinan | pag_Latn | Austronesian | Latin | 1,045 | 7,770 | 270,363 |
| Eastern Panjabi | pan_Guru | Indo-European | Gurmukhi | 10,857 | 44,440 | 1,821,511 |
| Papiamento | pap_Latn | Indo-European | Latin | 29,564 | 177,229 | 7,396,392 |
| Southern Pasto | pbt_Arab | Indo-European | Arabic | 31,854 | 107,563 | 27,623,486 |
| Western Persian | pes_Arab | Indo-European | Arabic | 6,995,368 | 24,998,370 | 6,061,794,870 |
| Plateau Malgasy | plt_Latn | Austronesian | Latin | 32,119 | 119,506 | 28,542,084 |
| Polish | pol_Latn | Indo-European | Latin | 14,492,239 | 60,362,860 | 10,994,239,010 |
| Portuguese | por_Latn | Indo-European | Latin | 8,033,406 | 26,058,040 | 4,639,089,792 |
| Dari | prs_Arab | Indo-European | Arabic | 421,097 | 2,101,038 | 399,037,437 |
| Ayacucho Quechua | quy_Latn | Quechuan | Latin | 1,248 | 10,038 | 322,112 |

| Languages | | | | Statistics | | |
|---|---|---|---|---|---|---|
| Lang. name | Code | Family | Script | #documents | #images | #tokens |
| Romanian | ron_Latn | Indo-European | Latin | 5,131,444 | 17,790,793 | 3,484,865,185 |
| Rundi | run_Latn | Atlantic-Congo | Latin | 17,798 | 55,060 | 8,140,230 |
| Russian | rus_Cyrl | Indo-European | Cyrillic | 15,753,144 | 68,786,134 | 18,196,141,357 |
| Sango | sag_Latn | Atlantic-Congo | Latin | 724 | 4,564 | 181,876 |
| Sicilian | scn_Latn | Indo-European | Latin | 27,388 | 164,772 | 17,535,500 |
| Sinhala | sin_Sinh | Indo-European | Sinhalese | 44,963 | 179,082 | 11,413,044 |
| Slovak | slk_Latn | Indo-European | Latin | 2,979,681 | 14,894,160 | 1,951,406,321 |
| Slovenian | slv_Latn | Indo-European | Latin | 1,456,026 | 7,106,291 | 928,101,642 |
| Samoan | smo_Latn | Austronesian | Latin | 11,024 | 62,358 | 11,672,900 |
| Shona | sna_Latn | Atlantic-Congo | Latin | 7,400 | 41,385 | 5,276,139 |
| Sindhi | snd_Arab | Indo-European | Arabic | 20,615 | 70,992 | 16,686,668 |
| Somali | som_Latn | Afro-Asiatic | Latin | 58,151 | 209,905 | 31,093,227 |
| Southern Sotho | sot_Latn | Atlantic-Congo | Latin | 7,474 | 41,714 | 5,876,842 |
| Spanish | spa_Latn | Indo-European | Latin | 22,218,630 | 76,372,709 | 13,882,047,139 |
| Sardinian | srd_Latn | Indo-European | Latin | 336,476 | 2,220,976 | 68,281,992 |
| Serbian | srp_Cyrl | Indo-European | Cyrillic | 593,332 | 2,251,042 | 394,477,097 |
| Sundanese | sun_Latn | Austronesian | Latin | 16,438 | 89,379 | 9,549,957 |
| Swedish | swe_Latn | Indo-European | Latin | 3,231,753 | 10,558,719 | 1,748,495,813 |
| Swahili | swh_Latn | Atlantic-Congo | Latin | 96,770 | 365,792 | 52,827,863 |
| Silesian | szl_Latn | Indo-European | Latin | 7,846 | 47,313 | 3,022,502 |
| Tamil | tam_Taml | Dravidian | Tamil | 30,202 | 149,837 | 4,234,345 |
| Tatar | tat_Cyrl | Turkic | Cyrillic | 34,489 | 133,014 | 22,255,423 |
| Telugu | tel_Telu | Dravidian | Telugu | 16,107 | 54,100 | 1,633,579 |
| Tajik | tgk_Cyrl | Turkic | Cyrillic | 119,383 | 395,470 | 87,519,228 |
| Tagalog | tgl_Latn | Austronesian | Latin | 140,922 | 628,210 | 95,285,900 |
| Thai | tha_Thai | Kra-Dai | Thai | 1,799,735 | 6,603,060 | 807,374,946 |
| Tigrinya | tir_Ethi | Afro-Asiatic | Ge'ez | 2,622 | 8,601 | 1,699,272 |
| Tok Pisin | tpi_Latn | Indo-European | Latin | 785 | 5,888 | 97,298 |
| Turkmen | tuk_Latn | Turkic | Latin | 12,372 | 54,002 | 9,650,172 |
| Turkish | tur_Latn | Turkic | Latin | 4,448,111 | 12,304,912 | 2,356,627,784 |
| Twi | twi_Latn | Atlantic-Congo | Latin | 286 | 2,041 | 78,227 |
| Uyghur | uig_Arab | Turkic | Arabic | 10,614 | 41,367 | 6,602,690 |
| Ukrainian | ukr_Cyrl | Indo-European | Cyrillic | 2,689,369 | 10,842,572 | 1,909,330,669 |
| Urdu | urd_Arab | Indo-European | Arabic | 403,245 | 1,224,175 | 236,356,788 |
| Northern Uzbek | uzn_Latn | Turkic | Latin | 113,772 | 581,861 | 81,808,833 |
| Venetian | vec_Latn | Indo-European | Latin | 122,390 | 763,029 | 24,081,966 |
| Vietnamese | vie_Latn | Viet-Muong | Latin | 12,296,989 | 46,339,341 | 11,462,111,787 |
| Wolof | wol_Latn | Atlantic-Congo | Latin | 2,152 | 9,351 | 367,848 |
| Xhosa | xho_Latn | Atlantic-Congo | Latin | 13,620 | 80,748 | 14,566,904 |
| Eastern Yiddish | ydd_Hebr | Indo-European | Hebrew | 12,275 | 56,421 | 17,078,751 |
| Yoruba | yor_Latn | Atlantic-Congo | Latin | 10,148 | 49,474 | 8,346,193 |
| Yue Chinese | yue_Hant | Sino-Tibetan | Hant | 28,478 | 172,592 | 21,579,579 |
| Chinese (Simplified) | zho_Hans | Sino-Tibetan | Hanzi | 8,326,440 | 29,575,591 | 5,199,137,981 |
| Chinese (Traditional) | zho_Hant | Sino-Tibetan | Hant | 3,796,336 | 15,514,804 | 2,617,463,485 |
| Standard Malay | zsm_Latn | Austronesian | Latin | 864,831 | 3,651,754 | 384,708,004 |
| Zulu | zul_Latn | Atlantic-Congo | Latin | 13,089 | 73,167 | 9,654,461 |

Table 1: Languages & Statistics

## A.2  HEURISTICS TO INCREASE THE QUALITY OF DOCUMENTS

We use a set of heuristics to improve the quality of the documents by discarding some text nodes. We first consider text nodes to be written in Latin scripts if more than 50% of the characters are Latin. In detail, we discard the text node if:

1. It is empty.

2. It contains fewer than 5 bytes for Latin scripts and fewer than 15 bytes for non-Latin scripts.

3. More than 30% of the characters are digits.

4. It contains more than one date.

5. It contains the sequence "lorem ipsum".

6. The ratio of non-alphabetic characters is superior to 0.33.

7. The symbols '{' or ''}' are in the text.

8. The symbols '$\geq$', '$\leq$', '>' or '<' are more than 2 times in the text.

9. "Follow us", "javascript", "copyright" or "©" are in the text.

10. The ratio of capitalized letters is superior to 0.2.

11. The text exactly matches with "comment", "facebook", "instagram", "twitter", "rss", "newsletter", "share" or "follow us".

12. A character is more than 33% of the total number of characters in the string.

We then also apply some filters to clean the text as much as possible:

1. Remove URLs from all documents.

2. Normalize consecutive special characters ('\t', '\n', '#', '/', '\$', ')', '(', '[', ']', '!', '?', '%', '<', '>') to keep only one.

Following previous steps, we keep the text node if it is superior to 5 bytes and we keep the final document if it is superior to 100 bytes.

## A.3 EXAMPLES OF DOCUMENTS

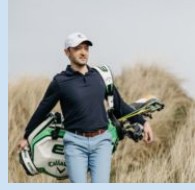 Autour des greens notre créativité est souvent mise à rude épreuve. En effet les bosses, la vitesse et la fermeté des greens, les obstacles à sauter, tous ces éléments nous poussent parfois à devoir modifier nos trajectoires de balles. Dans ces variations existe le lob shot ! Cette balle haute qui a pour objectif de survoler un obstacle et s'arrêter rapidement est souvent perçue comme un calvaire par les joueurs amateurs. Mais est ce si difficile ? Existe-t-il une manière de faire, « simple et répétitive », pour appréhender une première version de ce lob shot ? Je vais m'appuyer sur Jon Rahm, 7 ème cette année au Scrambling du PGA Tour* en dessous de 30m, pour vous apporter quelques explications pour améliorer ce domaine dans votre chipping.

Les premiers éléments à maitriser dans tous coups de golf sont les éléments de la posture ! Un stance ( position des pieds ) assez étroit. L'extérieur des pieds étant à l'intérieur de la largeur des épaules. Identique à la position classique de chipping. Le poids sur le pied avant = le droit pour les gauchers, le gauche pour les droitiers. Le club dans l'axe de l'aine et de l'avant bras comme indiqué par le trait vert. 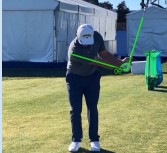 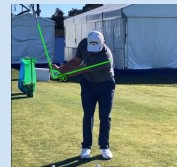

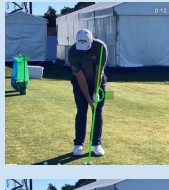 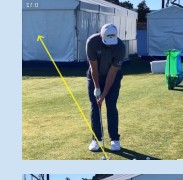 On voit également que la face de club est ouverte. Elle est en direction du ciel. Cette ouverture est effectué par une rotation de la face et non par une orientation de la face en avançant les mains vers l'avant, ce qui dans ce cas serait contre productif.

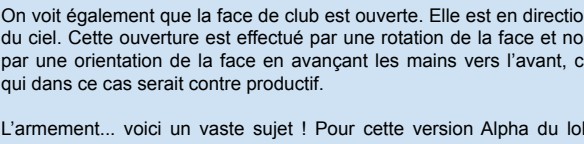

L'armement... voici un vaste sujet ! Pour cette version Alpha du lob shot, je vais vous demander d'envisager les choses ainsi. Si la face de club à l'adresse est ouverte le club en devient moins puissant. Exemple un F9 est moins puissant qu'un F5 ceci étant du, entre autre à l'ouverture de la face. Si le club est moins puissant et donc ici peu puissant, c'est un sand-wedge dont J. Rahm a ouvert la face, il faut pas mal d'amplitude même pour faire peu de distance. Si il faut de l'amplitude il faut, comme dans tout swing, se mettre à armer le club. L'armement dans cette version Alpha du lob shot n'est donc pas volontaire ! [...]

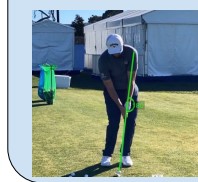 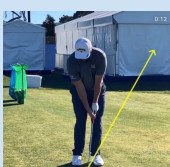

Figure 1: Example of a French document.

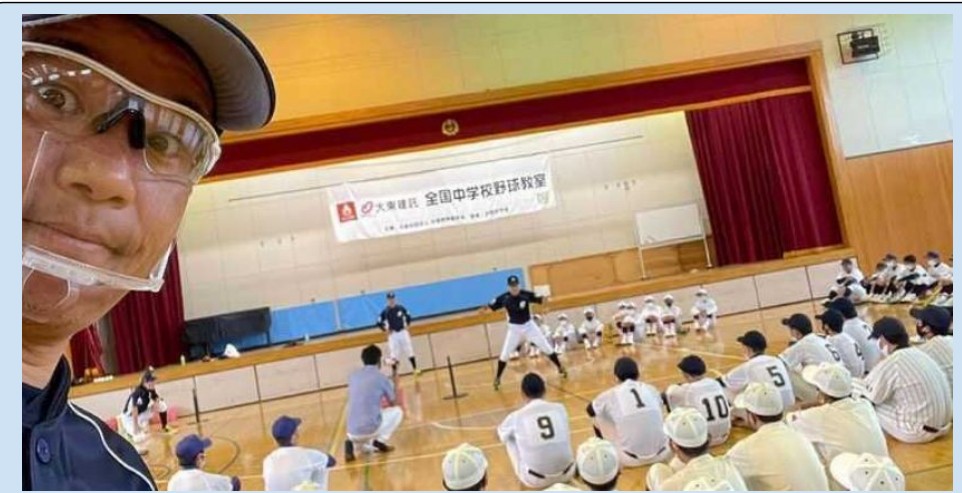

群馬県伊勢崎市でレジェンドたちと野球教室 ~! 本日、群馬県伊勢崎市にて野球教室でした〜。プロ野球OBクラブ更に「大東建託」さん主催！中学校の野球部の選手達へ熱血指導〜。

Figure 2: Example of a Japanese document.

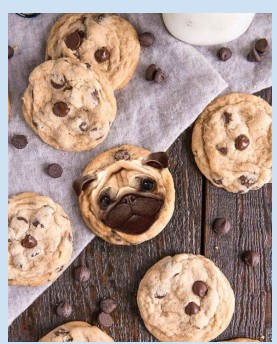

Собаки в еде! Необычный профиль в Instagram взорвал весь интернет. Данный аккаунт приглянется всем тем, кто не мыслит своей жизни без вкуснейшей еды и просто обожает братьев меньших, в особенности милых пёсиков. Только представьте себе, что у вас на тарелке лежит еда, но только в ней вы видите ещё и мордочку мопсика. Странно звучит, правда? Но вот кому-то эта идея пришла в голову и этот «кто-то» даже решил реализовать её. В Instagram в январе 2018 года появился весьма необычный профиль — @dogs_infood. В нём публикуются очень оригинальные и забавные иллюстрации, где изображена еда в тандеме с фотографиями собак.

Так что же можно там увидеть? Например, печенье с мордочкой мопса, веточка винограда со смешным французским бульдожкой, кренделёк с доберманом или шпиц в форме тефтельки. Это не только звучит забавно, но ещё и выглядит очень смешно. Кстати, любой желающий может прислать фотографию своего любимца автору профиля, и кто знает, может, следующий пост будет посвящён именно ему. [...]

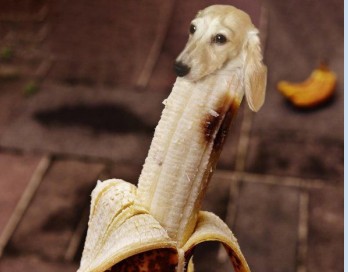

Figure 3: Example of a Russian document.

Nel mese di settembre c'è un altro evento sportivo che coinvolge soprattutto gli appassionati di corsa ed è il "Bibione is surprising run". É una gara internazionale di 10 miglia con percorsi che si intrecciano lungo il litorale toccando i punti più belli di Bibione. Anche per i meno allenati, è una buona occasione per far conciliare benessere fisico e salute. Ci sono tante proposte di strutture ricettive a Bibione che offrono pacchetti famiglia economici con la possibilità non solo di partecipare alla gara ma anche di fare un bel tuffo in mare. Il periodo di settembre è adatto per le famiglie con bambini: il mare è calmo e le giornate sono calde. Ritagliati un week-end last minute prima di tornare al lavoro e iniziare con la routine quotidiana. Di seguito sono elencati appartamenti confortevoli ed hotel economici che garantiscono risparmio e qualità al tuo soggiorno.

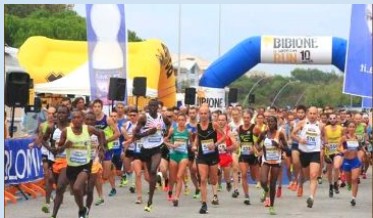

Rimani aggiornato sulle migliori offerte per Bibione. Residence con piscina - appartamento con barbecue e posto auto.

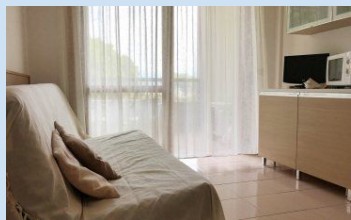

Figure 4: Example of an Italian document.

Nissan ចាប់ដៃគ្នាជាមួយ New Balance បញ្ចេញគំរូរថយន្តដ៏ពិសេសដែលមិនធ្លាប់មានពីមុនមក បែកធ្លាយរូបរាងឡ្មានរបស់ Tacoma ជំនាន់ថ្មី ចេញព័ត៌រូបប៉ាតង់ថ្មី មើលមកដូចកូន Tundra ៤ឆ្នាំទៀត Porsche នឹងឈប់ផលិត Macan ប្រើសាំង

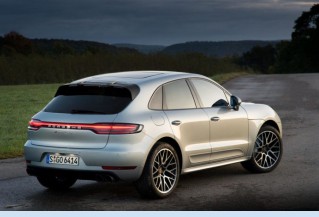 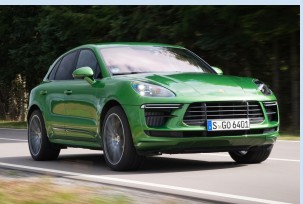 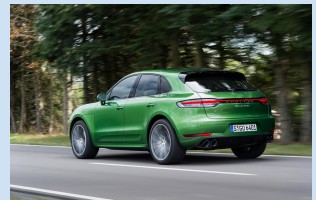

សមាជិក Blackpink សហការជាមួយ Porsche ឌីស្សាញម៉ូឌែលរថយន្តដ៏ពិសេសសម្រាប់ខ្លួនឯង

Figure 5: Example of a Khmer document.

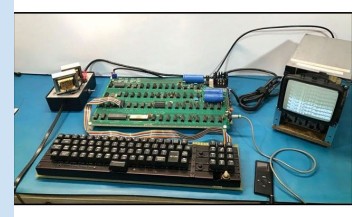

ایپل کا سب سے پہلا کمپیوٹر نیلامی کے لیے پیش

بوسٹن: ایپل کا سب سے پہلا مکمل طور پر فعال ایپل 1 کمپیوٹر نیلامی کے لیے پیش کر دیا گیا۔ میڈیا رپورٹ کے مطابق اس مشین، جس پر ایپل کے بانی اسٹیو جابز نے اپنے ہاتھوں سے نمبر ڈالے تھے، کے ساتھ وہ تمام چیزیں آئیں گی جو اس مشین کو چلانے کے لیے ضروری ہیں۔ فی الحال اس کمپیوٹر کی نیلامی کی بولی 2 لاکھ 41 ہزار 557 ڈالرز پر ہے جو 15 دسمبر کو ختم ہوجائے گی لیکن ایک اندازے کے مطابق اس کی حتمی بولی 3 لاکھ 75 ہزار ڈالرز تک جائے گی۔ 1976 میں متعارف کروایا جانے والا ایپل 1 اس ٹیک کمپنی کی سب سے پہلی شے تھی جو ایک اسمبلڈ سرکٹ بورڈ کے طور پر بیچی گئی تھی اس میں بنیادی چیزیں جیسے کہ کی بورڈ یا مانیٹر نہیں تھا۔ لیکن دیگر ایپل 1 کمپیوٹرز کے برعکس اس یونٹ کے فزیکل بورڈ میں کسی قسم کی کوئی تبدیلی نہیں کی گئی ہے اور اس کا نمونہ صاف اور بغیر کسی استعمال شدہ ہے۔ بوسٹن کے آکشن ہاؤس کے مطابق ایک تفصیلی ٹیسٹ میں اس سسٹم کو تقریباً آٹھ گھنٹے تک چلایا گیا جس میں کوئی خرابی سامنے نہیں آئی۔ تازہ ترین سلائیڈ شوز

Figure 6: Example of an Urdu document.

## A.4 TEXT-IMAGE SIMILARITY AND DOM TREE

As we rely on the DOM Tree to build the documents and the order of appearance of the nodes could differ from HTML rendering, we attempt to assess to what extent it is a relevant way of constructing a multimodal document. To do so, we rely on the results of the text-image joint filtering step where we compute the ranks of relevant text nodes (resp images) for each image. We plot the distribution of the closest most relevant node for each modality in Figures 7a and 7b. We notice that the most relevant node to either a text node or an image is their closest node in the DOM tree. The cumulative distribution function of the distribution of the closest node reaches 25% for nodes positioned between -5 and 5, which confirms the relevance of using the DOM tree to represent a document.

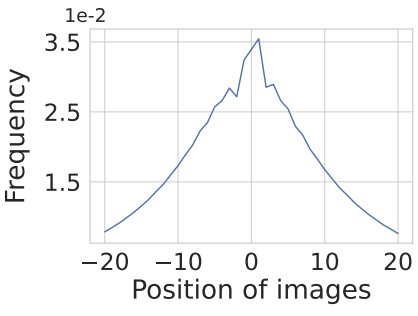

(a) Relative position in the document of relevant text nodes with respect to images.

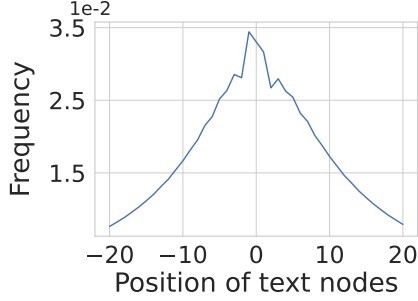

(b) Relative position in the document of relevant images with respect to text nodes.

Figure 7: Relative positions of most relevant images and text nodes with respect to the other modality.

## A.5 IMPLEMENTATION DETAILS

### A.5.1 TEXT DEDUPLICATION PARAMETERS

Following previous work, we near-deduplicate documents using MinHashLSH. We first vectorize the documents using HashingVectorizer from scikit-learn with 2,097,152 features computed on 4-grams and 5-grams within word boundaries. We then compute MinHashes from those vectors with 256

permutations and we finally run Locality Sensitive Hashing with a threshold Jaccard Similarity of 0.8 for finding near-duplicates.

### A.5.2 Removing Personal Identifiable Information

We used regular expressions to detect and remove PII in documents. More precisely, we used:

**email address**: `^[\w\.]+@[\w-]+\.[\w-]{2,4}$`

**phone number**: `^\+?\d{1,3}?[-.\s]?\(?\d{1,4}?\)?[-.\s]?\d{1,4}[-.\s]?\d{1,4}`
`[-.\s]?\d{1,9}$`

**credit card number**: `^(?:4[0-9]{12}(?:[0-9]{3})?|5[1-5][0-9]{14}|3[47][0-9]{13}|`
`3(?:0[0-5]|[68][0-9])[0-9]{11}|6(?:011|5[0-9]{2})[0-9]{12}|(?:2131|1800|35`
`\d{3})\d{11})$`

**IP address**: `^(?:25[0-5]|2[0-4][0-9]|1[0-9]{2}|[1-9][0-9]|\d)\.(?:25[0-5]|2[0-4]`
`[0-9]|1[0-9]{2}|[1-9][0-9]|\d)\.(?:25[0-5]|2[0-4][0-9]|1[0-9]{2}|[1-9][0-9]|`
`\d)\.(?:25[0-5]|2[0-4][0-9]|1[0-9]{2}|[1-9][0-9]|\d)$`

**passport number**: `^[A-Z0-9]{6,15}$`

For images, we detect faces in the images and distribute the bounding boxes coordinates. More precisely, all the images are resized to have a maximum of width and height of 256, keeping aspect ratio. The bounding boxes coordinates are therefore computed given this image size but can be extrapolated if images are downloaded in a higher resolution.

### A.5.3 Training implementation details

We train multilingual OpenFlamingo on mOSCAR and multilingual text-image pairs. We use a batch of size 64 for mOSCAR and 128 for captioning data, limiting the number of tokens to 256 for mOSCAR and 32 for captioning data. Similarly to Flamingo and OpenFlamingo, text tokens can only attend to the previous image in the sequence. To increase diversity in the training batch, we randomly reject 2/3 of the documents if they contain only one image. We limit the maximum number of images in a sequence to 8. We randomly sample 8 languages per batch and upsample low-resource languages. We train multilingual OpenFlamingo on 43 languages covering all the languages of the benchmarks we evaluate the models on (see Section A.5.4).

We use Gemma-2B as the underlying language model behind multilingual OpenFlamingo and CLIP ViT-L-14 as the image encoder. We add a cross-attention layer after each decoder layer. Following OpenFlamingo, we add the two special tokens `<image>` and `<|endofchunk|>`, whose embeddings were trained. Only the Perceiver Resampler, cross-attention layers and these two embeddings were trained; everything else remained frozen. During training, we apply a factor of 0.2 for the captioning data loss function.

We train the model using the Adam optimizer and a maximum learning rate of 1e-4. We use a constant learning rate scheduler with 1875 warm-up steps. We use 4 accumulation gradient steps to have an effective batch of size 256 for mOSCAR and 512 for captioning data. We train the model on 50M documents and 100M image-text pairs on 8 Nvidia A100 for 170h.

### A.5.4 Evaluation details

We evaluate on a set of eight benchmarks: xFlickr&CO, XM3600, xGQA, MaXM, MaRVL, XVNLI, Multi30k (Test2016 subset) and CoMMuTE; covering 5 different tasks and 43 languages. Details about the languages, the number of examples and the metric used can be found in Table 2. We used

| | Metric | #examples | Languages |
|---|---|---|---|
| xFlickr&CO | CideR | 2,000 | Chinese, English, German, Indonesian, Japanese, Russian, Spanish, Turkish |
| XM3600 | CideR | 3,600 | Arabic, Czech, Danish, German, Greek, English, Spanish, Farsi, Finnish, French, Hebrew, Hindi, Croatian, Hungarian, Indonesian, Italian, Japanese, Korean, Dutch, Norwegian, Poland, Portuguese, Romanian, Russian, Swedish, Telugu, Thai, Turkish, Ukrainian, Vietnamese, Chinese |
| xGQA | Accuracy | 9,666 | Bengali, German, English, Indonesian, Korean, Portuguese, Russian, Chinese |
| MaXM | Accuracy | $\sim 170$ | English, French, Hindi, Hebrew, Romanian, Thai, Chinese |
| MaRVL | Accuracy | $\sim 1,150$ | Indonesian, Swahili, Tamil, Turkish, Chinese |
| XVNLI | Accuracy | 1,164 | English, Arabic, Spanish, French, Russian |
| Multi30k | BLEU | 1,000 | French, German, Czech |
| CoMMuTE | Accuracy | 310 | Czech, French, German |

Table 2: Overview of the benchmarks used to evaluate our multilingual OpenFlamingo.

the *translate-test*[1] samples provided by the authors of the benchmarks if available. No translate test samples were provided for MaXM, so we translated the test set using the NLLB-600M distilled model. As no training set was available for MaXM, we use the few-shot examples from xGQA. Since we use Stanza tokenizers, we could not evaluate on all languages from XM3600 as 3 of them were not available. Filipino was also not into the list of mOSCAR languages, so we skip this language during evaluation. The CoMMuTE evaluation set involves choosing between two different translations of a same source text (one correct and one incorrect depending on an image provided to disambiguate the text). We use the lowest perplexity between the two translations as the model's prediction. We also use Multi30k training set as few-shot examples.

**Prompting** Following previous works, the zero-shot setting is composed of two few-shot examples without providing the images. The prompts we use for the different tasks are as follows:[2]

For captioning tasks, we use the prompt:

"`<image>Output:[Caption]<|endofchunk|><image>Output:`",

where `[Caption]` is replaced by the caption.

For visual question answering tasks, we use the prompt:

"`<image>Question: [Question] Short Answer: [Answer]`
`<|endofchunk|><image>Question: [Question] Short Answer:`",

where `[Question]` and `[Answer]` are replaced by the question and the answer respectively.

For multimodal machine translation tasks, we use the prompt:

"`<image>Sentence:'[Caption]'. Translation: [Translation]`
`<|endofchunk|><image>Output:`",

where `[Caption]` is replaced by the sentence to translate and `[Translation]` is replaced by its translation.

For MaRVL, we use the prompt:

"`<image> '[Statement]'. True of False? [Answer]<|endofchunk|><image>`
`'[Statement]'. True of False?`",

where `[Statement]` is replaced by the statement and `[Answer]` by the answer. We also concatenate the left and right image into a single image.

---

[1]Benchmark automatically translated into English.

[2]We show the prompts we used with one context example.

For XVNLI, we use the prompt:

```
"<image> `[Statement1]' – `[Statement2]'. entailment, neutral
or contradiction? Output: [Answer]<|endofchunk|><image>
`[Statement1]' – `[Statement2]'. entailment, neutral or
contradiction? Output:",
```

where `[Statement1]`, `[Statement2]` and `[Answer]` are replaced by XVNLI test data.

## A.6 DETAILED RESULTS

| | #shots | De | En | Es | Id | Ja | Ru | Tr | Zh |
|---|---|---|---|---|---|---|---|---|---|
| Multilingual OF *mOSCAR + caps.* | 0 | 26.93 | 29.64 | 14.07 | 32.04 | 2.87 | 18.07 | 4.23 | 7.40 |
| | 4 | 54.38 | 51.47 | 37.32 | 47.22 | 11.06 | 32.23 | 13.03 | 31.71 |
| | 8 | 55.09 | 56.75 | 34.99 | **51.60** | 15.03 | 34.17 | 13.63 | 33.90 |
| | 16 | **61.59** | **59.89** | **39.46** | 51.50 | **19.63** | **34.94** | **14.19** | **34.49** |
| Multilingual OF *captions only* | 0 | 16.72 | 24.57 | 3.80 | 10.82 | 2.82 | 8.20 | 2.79 | 6.82 |
| | 4 | 21.10 | 31.05 | 7.52 | 9.63 | 3.84 | 13.21 | 7.01 | 12.20 |
| | 8 | 32.56 | 35.73 | 13.35 | 15.85 | 5.96 | 18.13 | 6.97 | 15.47 |
| | 16 | 29.86 | 40.57 | 13.75 | 23.83 | 6.92 | 20.40 | 7.90 | 15.73 |

Table 3: Captioning results (CideR scores) on xFlickr&CO. **Bold** is best result.

| | #shots | Ar | Cs | Da | De | El | En | Es | Fa | Fi | Fr | He |
|---|---|---|---|---|---|---|---|---|---|---|---|---|
| Multi. OF *full* | 0 | 4.83 | 2.50 | 8.52 | 8.16 | 0.76 | 42.57 | 16.79 | 12.49 | 1.26 | 14.76 | 3.76 |
| | 4 | 22.74 | 6.42 | 33.73 | 24.29 | 2.32 | 77.98 | 37.81 | 31.94 | 6.78 | 39.79 | 15.51 |
| | 8 | 22.91 | 7.41 | 35.23 | **25.79** | 2.95 | 77.64 | 38.41 | **35.46** | 7.92 | 42.81 | 15.85 |
| | 16 | **23.47** | **8.14** | **35.96** | 25.47 | 2.58 | **78.18** | **39.18** | 31.44 | **8.42** | **43.77** | **16.08** |
| Multi. OF *Caps only* | 0 | 2.24 | 0.97 | 6.42 | 6.46 | 3.68 | 10.02 | 9.32 | 4.95 | 1.14 | 16.15 | 0.78 |
| | 4 | 5.36 | 1.36 | 13.11 | 11.82 | 7.78 | 35.52 | 19.96 | 9.62 | 1.86 | 22.48 | 2.29 |
| | 8 | 6.76 | 1.40 | 15.29 | 14.39 | 7.21 | 37.28 | 21.90 | 12.19 | 2.08 | 23.27 | 1.71 |
| | 16 | 6.25 | 2.29 | 17.96 | 15.11 | **7.64** | 48.03 | 25.39 | 9.21 | 2.10 | 30.16 | 2.72 |

| | #shots | Hi | Hr | Hu | Id | It | Ja | Ko | Nl | No | Pl | Pt |
|---|---|---|---|---|---|---|---|---|---|---|---|---|
| Multi. OF *full* | 0 | 2.79 | 2.00 | 1.51 | 9.96 | 11.53 | 0.92 | 0.58 | 16.11 | 8.31 | 3.94 | 13.37 |
| | 4 | 11.03 | 10.87 | 5.87 | 25.88 | 29.53 | 17.45 | 10.85 | 46.22 | 25.18 | 15.36 | 31.32 |
| | 8 | 11.61 | **12.00** | 6.91 | 29.68 | **29.34** | 20.13 | **12.01** | 47.58 | **27.08** | **17.80** | **33.29** |
| | 16 | **12.74** | 11.40 | 7.03 | 26.73 | 30.43 | 20.57 | 11.07 | **49.33** | 27.07 | 17.15 | 32.79 |
| Multi. OF *Caps only* | 0 | 2.29 | 0.97 | 3.51 | 2.98 | 7.96 | 1.85 | 1.05 | 4.88 | 5.78 | 0.92 | 9.79 |
| | 4 | 4.57 | 1.72 | 7.57 | 6.39 | 16.23 | 3.47 | 4.33 | 11.26 | 11.99 | 1.16 | 15.93 |
| | 8 | 5.94 | 2.17 | 7.83 | 9.93 | 15.40 | 7.93 | 5.34 | 11.87 | 13.79 | 1.38 | 17.50 |
| | 16 | 6.36 | 2.42 | **9.55** | 11.77 | 17.43 | 10.44 | 6.03 | 12.98 | 14.65 | 1.28 | 20.32 |

| | #shots | Ro | Ru | Sv | Te | Th | Tr | Uk | Vi | Zh |
|---|---|---|---|---|---|---|---|---|---|---|
| Multi. OF *full* | 0 | 1.84 | 4.72 | 11.09 | 0.88 | 5.49 | 2.86 | 2.08 | 11.34 | 3.29 |
| | 4 | 6.08 | 21.46 | 30.24 | 3.46 | 23.14 | 10.75 | 11.35 | 32.70 | 19.57 |
| | 8 | **7.10** | 21.78 | 30.26 | 3.76 | 25.17 | 12.83 | 12.26 | 35.86 | 20.11 |
| | 16 | 6.95 | **22.63** | **32.07** | **4.52** | **25.23** | **13.38** | **12.29** | **37.12** | **20.71** |
| Multi. OF *Caps only* | 0 | 2.24 | 1.93 | 4.55 | 0.67 | 2.34 | 2.68 | 0.80 | 8.55 | 2.70 |
| | 4 | 5.35 | 6.29 | 15.66 | 0.77 | 7.21 | 5.94 | 1.76 | 20.69 | 7.80 |
| | 8 | 5.18 | 7.58 | 14.01 | 1.00 | 6.81 | 8.90 | 2.73 | 23.05 | 8.99 |
| | 16 | 5.06 | 9.06 | 20.60 | 1.18 | 8.35 | 10.25 | 3.47 | 25.16 | 11.05 |

Table 4: Captioning results (CideR scores) on XM3600. **Bold** is best result.

| | #shots | Bn | De | En | Id | Ko | Pt | Ru | Zh |
|---|---|---|---|---|---|---|---|---|---|
| Multilingual OF *mOSCAR + caps.* | 0 | 22.76 | 25.72 | 34.24 | 26.68 | 26.89 | 26.73 | 25.28 | 27.32 |
| | 4 | 26.72 | 32.57 | 37.91 | 32.54 | 31.88 | 32.35 | 31.28 | 33.4 |
| | 8 | 28.07 | 35.15 | 39.44 | 35.14 | 32.94 | 35.59 | 33.58 | 34.04 |
| | 16 | **29.64** | **37.33** | **40.09** | **35.55** | **34.06** | **36.27** | **34.50** | **35.36** |
| Multilingual OF *captions only* | 0 | 10.54 | 6.51 | 10.43 | 7.74 | 7.50 | 7.79 | 8.62 | 9.84 |
| | 4 | 12.54 | 11.90 | 15.78 | 13.95 | 13.70 | 12.01 | 12.73 | 15.03 |
| | 8 | 11.62 | 11.70 | 17.29 | 13.86 | 12.85 | 11.60 | 12.65 | 15.35 |
| | 16 | 9.77 | 11.86 | 18.37 | 13.24 | 12.48 | 11.25 | 11.24 | 14.33 |
| *Translate Test* | | | | | | | | | |
| OF-3B MPT | 0 | 18.64 | 18.67 | - | 18.36 | 17.54 | 19.21 | 18.88 | 17.11 |
| | 4 | 23.23 | 23.40 | - | 22.95 | 22.46 | 23.52 | 22.41 | 22.85 |
| | 8 | 28.22 | 29.44 | - | 28.21 | 27.67 | 29.58 | 28.21 | 28.63 |
| | 16 | 31.31 | 32.58 | - | 31.82 | 31.42 | 32.74 | 31.62 | 31.22 |
| Multilingual OF *mOSCAR + caps.* | 0 | 30.41 | 32.1 | - | 29.35 | 29.99 | 31.39 | 29.06 | 28.81 |
| | 4 | 34.89 | 36.32 | - | 35.50 | 35.64 | 36.84 | 35.05 | 34.60 |
| | 8 | 35.95 | 37.65 | - | 36.78 | 37.14 | 37.81 | 36.17 | 35.98 |
| | 16 | **36.78** | **38.78** | - | **37.52** | **37.73** | **38.68** | **37.91** | **36.84** |

Table 5: VQA results on xGQA. **Bold** is best result.

| | #shots | En | Fr | Hi | He | Ro | Th | Zh |
|---|---|---|---|---|---|---|---|---|
| Multi. OF *mOSCAR + caps* | 0 | 36.58 | 28.03 | 20.38 | 18.21 | 15.49 | 24.25 | 13.36 |
| | 4 | 38.13 | 30.03 | 23.08 | 21.43 | 17.61 | 31.72 | 22.02 |
| | 8 | 38.52 | 29.55 | 24.62 | 20.00 | 17.61 | **25.27** | **23.83** |
| | 16 | 35.80 | **31.82** | **25.00** | **23.93** | 19.01 | **33.96** | 22.74 |
| Multi. OF *captions only* | 0 | 9.73 | 0.38 | 7.69 | 1.43 | 0.00 | 5.22 | 3.61 |
| | 4 | 9.34 | 2.65 | 5.00 | 2.50 | 0.00 | 5.60 | 3.97 |
| | 8 | 9.34 | 1.89 | 8.08 | 5.00 | 1.06 | 3.36 | 5.42 |
| | 16 | 8.56 | 1.14 | 5.00 | 8.21 | 0.35 | 3.36 | 7.58 |
| *Translate test* | | | | | | | | |
| OF-3B MPT | 0 | - | 12.50 | 22.31 | 0.36 | 10.92 | 0.00 | 0.00 |
| | 4 | - | 10.98 | 25.38 | 0.36 | 10.21 | 0.00 | 0.00 |
| | 8 | - | 10.98 | 27.31 | 0.36 | 11.27 | 0.00 | 0.00 |
| | 16 | - | 13.26 | 26.54 | **1.07** | 13.38 | 0.00 | 0.00 |
| Multi. OF *mOSCAR + caps* | 0 | - | **18.18** | 28.08 | 0.00 | 13.73 | 0.00 | **0.36** |
| | 4 | - | 15.91 | 30.38 | 0.36 | 12.68 | 0.00 | 0.00 |
| | 8 | - | 15.15 | 30.77 | 0.00 | 14.79 | 0.00 | 0.00 |
| | 16 | - | 15.91 | **35.77** | 0.36 | **16.90** | 0.00 | 0.00 |

Table 6: VQA results on MaXM. **Bold** is best result.

|  | #shots | Id | Sw | Ta | Tr | Zh |
|---|---|---|---|---|---|---|
| Random chance |  | 50.00 | 50.00 | 50.00 | 50.00 | 50.00 |
| Multilingual OF *mOSCAR + caps* | 0 | 50.09 | 49.46 | 49.60 | 49.83 | 48.81 |
|  | 4 | 49.91 | 48.19 | 49.68 | 50.42 | 50.00 |
|  | 8 | **53.55** | **50.72** | 49.76 | **51.78** | **51.58** |
|  | 16 | 48.94 | 49.82 | 49.20 | 50.25 | 50.99 |
| Multilingual OF *captions only* | 0 | 51.33 | 49.01 | 49.52 | 49.83 | 49.70 |
|  | 4 | 49.73 | 49.64 | 49.19 | 49.41 | 49.70 |
|  | 8 | 49.91 | 49.10 | 49.60 | 49.75 | 49.90 |
|  | 16 | 50.09 | 49.73 | 49.60 | 49.75 | 49.80 |
| *Translate test* |  |  |  |  |  |  |
| OF-3B MPT | 0 | 50.00 | 49.37 | 49.76 | 49.83 | 49.80 |
|  | 4 | 50.00 | 49.64 | 49.52 | 49.75 | 49.60 |
|  | 8 | 49.82 | 49.46 | 49.28 | 50.08 | 49.90 |
|  | 16 | 50.00 | 49.37 | 49.44 | 50.00 | 49.80 |
| Multilingual OF *mOSCAR + caps* | 0 | 49.07 | 49.79 | 49.52 | 50.34 | 49.60 |
|  | 4 | 49.99 | 49.79 | 48.23 | 49.75 | 49.76 |
|  | 8 | 50.00 | 48.92 | **50.64** | 50.42 | 48.90 |
|  | 16 | 49.84 | 50.00 | 50.24 | 48.90 | 49.75 |

Table 7: Classification results on MaRVL. **Bold** is best result.

|  | #shots | Ar | En | Es | Fr | Ru |
|---|---|---|---|---|---|---|
| Random chance |  | 33.33 | 33.33 | 33.33 | 33.33 | 33.33 |
| Multilingual OF. *mOSCAR + caps.* | 0 | 33.51 | 34.62 | 33.08 | 34.02 | 34.19 |
|  | 4 | 33.08 | 33.59 | 33.42 | 34.45 | 35.82 |
|  | 8 | **35.91** | **38.75** | **35.14** | **36.08** | **37.11** |
|  | 16 | 34.11 | 36.60 | 33.93 | 34.54 | 35.05 |
| Multilingual OF. *captions only* | 0 | 35.48 | 34.02 | 33.51 | 34.45 | 31.36 |
|  | 4 | 32.04 | 31.79 | 32.73 | 32.22 | 31.44 |
|  | 8 | 34.02 | 33.76 | 32.04 | 35.57 | 33.16 |
|  | 16 | 32.04 | 32.99 | 33.76 | 33.17 | 31.53 |
| *Translate test* |  |  |  |  |  |  |
| OF-3B MPT | 0 | 32.65 | - | 31.01 | 31.44 | 35.82 |
|  | 4 | 36.25 | - | 35.82 | 35.57 | 35.65 |
|  | 8 | 31.27 | - | 31.10 | 31.10 | 31.70 |
|  | 16 | 33.68 | - | 33.25 | 32.99 | 33.25 |
| Multilingual OF. *mOSCAR + caps.* | 0 | 34.88 | - | 34.88 | 34.54 | 34.36 |
|  | 4 | 36.25 | - | 36.17 | 35.91 | 36.08 |
|  | 8 | **39.60** | - | **39.52** | **40.29** | **39.35** |
|  | 16 | 37.54 | - | 37.89 | 37.46 | 39.00 |

Table 8: Classification results on XVNLI. **Bold** is best result.

| | #shots | Cs | De | Fr |
|---|---|---|---|---|
| Multi. OF *full* | 0 | 2.82 | 28.45 | 37.47 |
| | 4 | 3.12 | 29.20 | 37.49 |
| | 8 | 3.14 | **29.62** | 37.99 |
| | 16 | **3.34** | 29.41 | **38.79** |
| Multi. OF *caps. only* | 0 | 0.00 | 0.00 | 0.00 |
| | 4 | 0.00 | 0.00 | 0.00 |
| | 8 | 0.00 | 0.00 | 0.03 |
| | 16 | 0.00 | 0.40 | 1.82 |

Table 9: En→X translation results on Multi30k. **Bold** is best result.

| | #shots | Cs | De | Fr |
|---|---|---|---|---|
| Multi. OF *full* | 0 | 56.49 | **65.67** | 67.86 |
| | 4 | 57.47 | 64.00 | **68.18** |
| | 8 | 58.44 | 64.33 | 67.86 |
| | 16 | 58.11 | 62.67 | 66.23 |
| Multi. OF *caps. only* | 0 | 58.12 | 61.67 | 64.29 |
| | 4 | **59.09** | 61.00 | 63.31 |
| | 8 | **59.09** | 59.34 | 64.29 |
| | 16 | 58.12 | 58.67 | 63.96 |

Table 10: En→X CoMMuTE results. **Bold** is best result.

## A.7 Comparison with state-of-the-art mLLMs

| | # shots | xFlickR&CO | XM3600 | xGQA | MaXM | MaRVL | XVNLI | Multi30k | CoMMuTE |
|---|---|---|---|---|---|---|---|---|---|
| InternVL2 4B | 0 | 16.21 | 7.02 | 12.38 | 6.35 | 53.14 | 33.85 | 26.99 | 66.93 |
| | 4 | 24.89 | 9.53 | 26.05 | 14.72 | 54.22 | 35.72 | 26.68 | 64.22 |
| PaliGemma 3B | 0 | 28.28 | **24.49** | **42.68** | **33.42** | 51.48 | 39.36 | 17.98 | 62.78 |
| Idefics2 8B | 0 | 27.11 | 15.94 | 22.53 | 28.99 | **63.18** | **50.33** | **30.19** | **67.13** |
| Llava-NeXT 8B | 0 | 23.67 | 14.70 | 25.48 | 15.17 | 60.50 | 45.40 | 29.40 | 66.37 |
| Multi. OF 3B (*ours*) | 0 | 16.91 | 7.45 | 26.95 | 22.23 | 49.56 | 33.88 | 22.91 | 63.34 |
| | 4 | **34.80** | 22.18 | 32.23 | 26.33 | 49.64 | 34.07 | 23.27 | 63.22 |

Table 11: Results averaged across languages. **Bold** is best result.

We computed the results for different state-of-the-art models of similar sizes as multilingual Open Flamingo namely: (1) InternVL2-4B[3] (2) PaliGemma[4] (3) Idefics2-8B[5] and (3) Llava-NeXT 8B[6]. InternVL2 and PaliGemma are trained on multilingual and multimodal data while Llava-NeXT and Idefics2 are trained on English multimodal datasets.

Table 11 shows results averaged across languages for different state-of-the-art mLLMs of sizes from 3b to 8B. These results highlights multiple things: (1) getting results significantly better than random (MaRVL and XVNLI) requires instruction-tuning data as Idefics2 and Llava-NeXT were both trained on instruction-tuning multimodal datasets (2) English-only still gets decent results on multilingual benchmarks despite not having been trained on multilingual and multimodal data, probably due to their underlying LLM being multilingual (3) multilingual Open Flamingo (trained on mOSCAR and captions) gets superior results to InternVL2-4B on VQA benchmarks and captioning benchmarks but inferior to PaliGemma-3B mainly due to the fact that it was trained on much less data and the quality of the captions used to train multilingual Open Flamingo may not be as good as the WebLI dataset used to train PaliGemma.

---

[3]`OpenGVLab/InternVL2-4B`
[4]`google/paligemma-3b-pt-224`
[5]`HuggingFaceM4/idefics2-8b`
[6]`llava-hf/llama3-llava-next-8b-hf`

A.8   DATASHEET FOR MOSCAR

**Motivation**

**For what purpose was the dataset created?** *Was there a specific task in mind? Was there a specific gap that needed to be filled? Please provide a description.*

Existing large-scale interleaved image-text datasets available are English-only. We create a similar dataset but we cover 163 languages in order to train multilingual multimodal language models.

**Composition**

**What do the instances that comprise the dataset represent (e.g., documents, photos, people, countries)?** *Are there multiple types of instances (e.g., movies, users, and ratings; people and interactions between them; nodes and edges)? Please provide a description.*

The instances represent web documents with raw text interleaved with images.

**How many instances are there in total (of each type, if appropriate)?**

There are approximately 303 million instances (documents) in the dataset.

**Does the dataset contain all possible instances or is it a sample (not necessarily random) of instances from a larger set?** *If the dataset is a sample, then what is the larger set? Is the sample representative of the larger set (e.g., geographic coverage)? If so, please describe how this representativeness was validated/verified. If it is not representative of the larger set, please describe why not (e.g., to cover a more diverse range of instances, because instances were withheld or unavailable).*

The dataset is complete. Instances were filtered (and therefore not included in the dataset) because of not meeting certain criteria, including quality, spam filters, NSFW filters, as described in the article.

**What data does each instance consist of?  "Raw" data (e.g., unprocessed text or images) or features?** *In either case, please provide a description.*

The instances of the dataset are composed of two lists and one dictionary. The first element is a list of URLs and the index of the related images in the document. The second one is a list of raw text with its index in the document. The last one is a dictionary of metadata containing the order of the indexes to build the document, the URL of the document and the language assigned to the document.

**Is there a label or target associated with each instance?** *If so, please provide a description.*

No.

**Is any information missing from individual instances?** *If so, please provide a description, explaining why this information is missing (e.g., because it was unavailable). This does not include intentionally removed information, but might include, e.g., redacted text.*

No information we are aware of.

**Are relationships between individual instances made explicit (e.g., users' movie ratings, social network links)?** *If so, please describe how these relationships are made explicit.*

Individual instances (i.e. documents) are independent from each other.

**Are there recommended data splits (e.g., training, development/validation, testing)?** *If so, please provide a description of these splits, explaining the rationale behind them.*

There are no recommended data splits as mOSCAR is a pretraining dataset.

**Are there any errors, sources of noise, or redundancies in the dataset?** *If so, please provide a description.*

mOSCAR is a web-crawled large-scale so it is noisy by construction. We applied a series of steps to maximise the quality of the dataset and to remove near-duplicates from the dataset, but we cannot be sure that all duplicates have been removed.

**Is the dataset self-contained, or does it link to or otherwise rely on external resources (e.g., websites, tweets, other datasets)?** *If it links to or relies on external resources, a) are there guarantees that they will exist, and remain constant, over time; b) are there official archival versions of the complete dataset (i.e., including the external resources as they existed at the time the dataset was created); c) are there any restrictions (e.g., licenses, fees) associated with any of the external resources that might apply to a future user? Please provide descriptions of all external resources and any restrictions associated with them, as well as links or other access points, as appropriate.*

The dataset is almost self-contained. Users are required to collect the images from the set of URLs, as we cannot share the images directly ourselves. The dataset will therefore not remain constant over time as some images can change or be deleted.

**Does the dataset contain data that might be considered confidential (e.g., data that is protected by legal privilege or by doctor-patient confidentiality, data that includes the content of individuals non-public communications)?** *If so, please provide a description.*

We did not notice such data when inspecting a subsample manually. However, given the scale of the dataset, it is possible that it includes personal information. We respected robots.txt instructions when collecting data to limit this presence of PII.

**Does the dataset contain data that, if viewed directly, might be offensive, insulting, threatening, or might otherwise cause anxiety?** *If so, please describe why.*

We did our best to remove NSFW content from the text or the images. Documents with NSFW content were removed from the dataset. We however did not do any analysis of toxicity as this is very challenging in such a multilingual dataset. At such scale, it is therefore possible that users could find some offensive content.

**Does the dataset relate to people?** *If not, you may skip the remaining questions in this section.*

No.

**Does the dataset identify any subpopulations (e.g., by age, gender)?** *If so, please describe how these subpopulations are identified and provide a description of their respective distributions within the dataset.*

Indirectly, as the data is crawled from the web, it conveys the representation of populations widespread on the internet.

**Is it possible to identify individuals (i.e., one or more natural persons), either directly or indirectly (i.e., in combination with other data) from the dataset?** *If so, please describe how.*

It is possible to identify public figures within the dataset. It might also be possible to identify individuals if they are present on the internet as the dataset is web-crawled. However, the text is raw text and no identifying labels were added to the dataset.

**Does the dataset contain data that might be considered sensitive in any way (e.g., data that reveals racial or ethnic origins, sexual orientations, religious beliefs, political opinions or union memberships, or locations; financial or health data; biometric or genetic data; forms of government identification, such as social security numbers; criminal history)?** *If so, please provide a description.*

Again, as it is a large-scale web-crawled dataset, it might contain sensitive data.

**Collection Process**

**How was the data associated with each instance acquired?** *Was the data directly observable (e.g., raw text, movie ratings), reported by subjects (e.g., survey responses), or indirectly inferred/derived*

*from other data (e.g., part-of-speech tags, model-based guesses for age or language)? If data was reported by subjects or indirectly inferred/derived from other data, was the data validated/verified? If so, please describe how.*

The data was directly observable. It was raw text from webpages.

**What mechanisms or procedures were used to collect the data (e.g., hardware apparatus or sensor, manual human curation, software program, software API)?** *How were these mechanisms or procedures validated?*

We did not any procedures to collect the text data as it was extracted from CommonCrawl. We collected the images using a modified version of img2dataset that stores the robots.txt instructions from websites and follows them strictly. We additionally did not collect the images if CCBot agent was disallowed as the data is originally from CommonCrawl.

**If the dataset is a sample from a larger set, what was the sampling strategy (e.g., deterministic, probabilistic with specific sampling probabilities)?**

The dataset is from the larger subset CommonCrawl. We detailed the filtering procedures in the core of the paper.

**Who was involved in the data collection process (e.g., students, crowdworkers, contractors) and how were they compensated (e.g., how much were crowdworkers paid)?**

Only authors were involved in the data collection process.

**Over what timeframe was the data collected? Does this timeframe match the creation timeframe of the data associated with the instances (e.g., recent crawl of old news articles)?** *If not, please describe the timeframe in which the data associated with the instances was created.*

We collect data from three CommonCrawl dumps of 2023. The collection process spanned from January 2024 to March 2024.

**Were any ethical review processes conducted (e.g., by an institutional review board)?** *If so, please provide a description of these review processes, including the outcomes, as well as a link or other access point to any supporting documentation.*

No.

**Does the dataset relate to people?** *If not, you may skip the remaining questions in this section.*

It can indirectly relate to people as it is a large-scale web-crawled dataset.

**Did you collect the data from the individuals in question directly, or obtain it via third parties or other sources (e.g., websites)?**

N/A

**Were the individuals in question notified about the data collection?** *If so, please describe (or show with screenshots or other information) how notice was provided, and provide a link or other access point to, or otherwise reproduce, the exact language of the notification itself.*

N/A

**Did the individuals in question consent to the collection and use of their data?** *If so, please describe (or show with screenshots or other information) how consent was requested and provided, and provide a link or other access point to, or otherwise reproduce, the exact language to which the individuals consented.*

N/A

**If consent was obtained, were the consenting individuals provided with a mechanism to revoke their consent in the future or for certain uses?** *If so, please provide a description, as well as a link or other access point to the mechanism (if appropriate).*

We cannot obtain consent of all website owners. We can however remove the webpage or a specific image if a request is made.

**Has an analysis of the potential impact of the dataset and its use on data subjects (e.g., a data protection impact analysis) been conducted?** *If so, please provide a description of this analysis, including the outcomes, as well as a link or other access point to any supporting documentation.*

N/A

**Preprocessing/cleaning/labeling**

**Was any preprocessing/cleaning/labeling of the data done (e.g., discretization or bucketing, tokenization, part-of-speech tagging, SIFT feature extraction, removal of instances, processing of missing values)?** *If so, please provide a description. If not, you may skip the remainder of the questions in this section.*

We described all the processing and cleaning steps in the core of the paper.

**Was the "raw" data saved in addition to the preprocessed/cleaned/labeled data (e.g., to support unanticipated future uses)?** *If so, please provide a link or other access point to the "raw" data.*

The raw data is available from CommonCrawl. We only release the processed data.

**Is the software used to preprocess/clean/label the instances available?** *If so, please provide a link or other access point.*

We only use open-source tools to process the data except Safer, a proprietary child sexual abuse material detector to remove CSAM from the dataset.

**Uses**

**Has the dataset been used for any tasks already?** *If so, please provide a description.*

We did some experiments we reported in the core of the paper.

**Is there a repository that links to any or all papers or systems that use the dataset?** *If so, please provide a link or other access point.*

We will release code and models we used in the paper.

**What (other) tasks could the dataset be used for?**

It is a pretraining dataset.

**Is there anything about the composition of the dataset or the way it was collected and preprocessed/cleaned/labeled that might impact future uses?** *For example, is there anything that a future user might need to know to avoid uses that could result in unfair treatment of individuals or groups (e.g., stereotyping, quality of service issues) or other undesirable harms (e.g., financial harms, legal risks) If so, please provide a description. Is there anything a future user could do to mitigate these undesirable harms?*

Users could develop methods to mitigate biases and toxicity in such a large-scale multilingual dataset.

**Are there tasks for which the dataset should not be used?** *If so, please provide a description.*

No task we are aware of.

**Distribution**

**Will the dataset be distributed to third parties outside of the entity (e.g., company, institution, organization) on behalf of which the dataset was created?** *If so, please provide a description.*

Yes, the dataset will be publicly available.

**How will the dataset will be distributed (e.g., tarball on website, API, GitHub)** *Does the dataset have a digital object identifier (DOI)?*

The dataset will be distributed and maintained on HuggingFace.

**When will the dataset be distributed?**

The dataset is already distributed on the HuggingFace hub.

**Will the dataset be distributed under a copyright or other intellectual property (IP) license, and/or under applicable terms of use (ToU)?** *If so, please describe this license and/or ToU, and provide a link or other access point to, or otherwise reproduce, any relevant licensing terms or ToU, as well as any fees associated with these restrictions.*

The dataset will be distributed under the Creative Commons Attribution 4.0 International (CC-BY-4.0) license.

**Have any third parties imposed IP-based or other restrictions on the data associated with the instances?** *If so, please describe these restrictions, and provide a link or other access point to, or otherwise reproduce, any relevant licensing terms, as well as any fees associated with these restrictions.*

It is possible that instructions from websites' owners to allow the collection of the data change over time. People must follow these instructions when they collect images and must not collect them if the owner puts restrictions.

**Do any export controls or other regulatory restrictions apply to the dataset or to individual instances?** *If so, please describe these restrictions, and provide a link or other access point to, or otherwise reproduce, any supporting documentation.*

No.

**Maintenance**

**Who will be supporting/hosting/maintaining the dataset?**

We will host the dataset on the HuggingFace hub.

**How can the owner/curator/manager of the dataset be contacted (e.g., email address)?**

Email address of the first author is provided.

**Is there an erratum?** *If so, please provide a link or other access point.*

No.

**Will the dataset be updated (e.g., to correct labeling errors, add new instances, delete instances)?** *If so, please describe how often, by whom, and how updates will be communicated to users (e.g., mailing list, GitHub)?*

There are no current plans to update the dataset, unless specific requests are made, such as removing certain image URLs. However, we do not exclude providing an updated version in the future.

**If the dataset relates to people, are there applicable limits on the retention of the data associated with the instances (e.g., were individuals in question told that their data would be retained for a fixed period of time and then deleted)?** *If so, please describe these limits and explain how they will be enforced.*

At such scale, it is unfeasible to contact all people having data in the dataset.

**Will older versions of the dataset continue to be supported/hosted/maintained?** *If so, please describe how. If not, please describe how its obsolescence will be communicated to users.*

The dataset will continue to be hosted on the HuggingFace hub.

**If others want to extend/augment/build on/contribute to the dataset, is there a mechanism for them to do so?** *If so, please provide a description. Will these contributions be validated/verified? If so, please describe how. If not, why not? Is there a process for communicating/distributing these contributions to other users? If so, please provide a description.*

We will verify any contributions made to the dataset. To contribute please contact the authors of mOSCAR.

**Authors Statement**  mOSCAR is released under the CC-BY 4.0 license. Users should respect its terms of use. We bear all responsability in case of violation of rights.