# OpenReview forum: "mOSCAR: A Large-scale Multilingual and Multimodal Document-Level Corpus"
_ICLR.cc/2025/Conference — Submitted to ICLR 2025_

### Official Review · Reviewer_Fgsw · 2024-11-04

**Soundness:** 3
**Presentation:** 4
**Contribution:** 3
**Rating:** 6
**Confidence:** 4

**Summary:**

This paper constructs a fully open-sourced multilingual multimodal datasets for interleaved text-image training. The dataset, namely mOSCAR, covers 163 languages, 303M documents, 200B tokens, and 1.15B images. The authors compare the performance of one model trained on a subset of mOSCAR and caption data with another trained on caption data only, and find the model trained on mOSCAR exhibits better in-context learning abilities.

**Strengths:**

1. The paper collects a large amount of data of high quality for multilingual multimodal training and open-source it. The workload and effort is significantly high and the dataset will be useful to the community.
2. The data processing pipeline is comprehensive and containing standard steps (e.g., deduplication, language filtering, PII removal, etc.), resulting in high-quality data.
3. The analysis shows mOCSCAR matches the quality of the existing interleaved English-only datasets in terms of quality and diversity

**Weaknesses:**

1. I appreciate this line of work, but the contribution beyond engineering effort is unclear.
2. The baselines in performance comparison needs to be further justified (see questions).
3.It is unclear how much the English-only performance drop when training on this multilingual dataset, where the author should conduct an ablation study and present the model performance on common English-only dataset such as MMMU, MME, etc. If there is an obvious decrease in English performance, the author should discuss how to mitigate this effect and in what situation the users should choose mOSCAR and combine with other caption data and with what ratio.

**Questions:**

1. Could the authors clarify their baselines? In L399-L401, the authors said they trained 3 models on: (1) 50M mOSCAR + 100M image-text pairs; (2) 300M image-text pairs; (3) 35M WIT + 70M text-image pairs. While in Table 5 there is one new setting with 35M mOSCAR + 70M text-image pairs. It would be helpful for the authors to clarify their model training settings in every caption to avoid confusion.
2. Could the author explain why WIT performs better than mOSCAR with the same performance score across different number of shots on Multi30K in Table 5?
3. What is the trade-off of using English-only dataset and mOSCAR? Could the author evaluate the models on English-only benchmarks as well to indicate in what situation the user should choose mOCSAR and at what ratio?
4. I also wonder if there is any trade-off between enhancing the VLM's ability on few-shot learning situation v.s. other capabilities such as general perception, captioning, VQA, etc. Could the authors design analysis on this?
5. With what license the authors plan to release the data?

**Details Of Ethics Concerns:**

No particular concern.

---

> ### Author Response · Authors · 2024-11-19
>
> Thank you very much for your review. Please see our answers to your weaknesses below and our answers to your questions in the next comment.
>
> **W1/ Contributions**
>
> Our contributions include the dataset design and the method to build such a large-scale multilingual and multimodal dataset, the first of its kind in the multilingual setting. Furthermore, our experimental results show that the findings from [1], [2], [3] stating that interleaved data introduces a boost in few-show learning holds in the multilingual setting, as highlighted by Table 4 in Section 5.
>
> **W2/ Baselines**
>
> Please see our answers to your question 1/ and 2/ regarding the clarification of our baselines (next comment).
>
> **W3/ Performances of the multilingual model on English benchmarks**
>
> To evaluate the impact on English benchmarks when training on mOSCAR (i.e. multilingual) we run additional experiments (see new version of the PDF). First of all, we retrained an OpenFlamingo on the English subset of mOSCAR and English captions (“English OF” trained on 50M documents and 100M captions) and compared it to multilingual OpenFlamingo (multilingual OF) trained on the 43 languages (same amount of data seen during training). Below is a comparison between both models (multilingual OF & English OF) on the English subsets of the benchmarks we used in the paper: (same table can be found in the new version of the paper, see new pdf).
>
> | 	  | # shots   | xFlickR&CO | XM3600 | xGQA | MaXM | XVNLI |
> |-----------------|-----------------|-----------------|-----------------|-----------------|-----------------|-----------------|
> | Multilingual OF (**mOSCAR** + captions) | 0 | 29.64 | 42.57 | 34.24 | 36.58 | 34.62 |
> | Multilingual OF (**mOSCAR** + captions) | 4 | 51.47 | 77.98 | 37.91 | 38.13 | 33.59 |
> | Multilingual OF (**mOSCAR** + captions) | 8 | 56.75 | 77.64 | 39.44 | 38.52 | 38.75 |
> | Multilingual OF (**mOSCAR** + captions) | 16 | 59.89 | 78.18 | 40.09 | 35.80 | 36.60 |
> | English OF (English subset of **mOSCAR** + English captions) | 0 | 32.70 | 43.75 | 34.71 | 36.19 | 35.82 |
> | English OF (English subset of **mOSCAR** + English captions) | 4 | 51.39 | 75.33 | 37.48 | 37.96 | 34.88 |
> | English OF (English subset of **mOSCAR** + English captions) | 8 | 51.44 | 77.73 | 39.64 | 38.35 | 36.86 |
> | English OF (English subset of **mOSCAR** + English captions) | 16 | 59.24 | 78.38 | 40.36 | 37.35 | 37.11 |
>
> We observe very similar results on English evaluation sets when training on multiple languages of mOSCAR in comparison to the model trained on the English-only subset.
>
> We also evaluated these models on MMMU & MME, results can be found in the Table below. The results on these benchmarks are quite similar, meaning that the impact of the multilingual data vs English-only is neglectable. Results are low for all models because Flamingo-like architectures are not well adapted to perform on these benchmarks. They are however cheaper to train than state-of-the-art vision language models.
>
> | 	  | MMMU   | MME |
> |-----------------|-----------------|-----------------|
> | Multilingual OF (**mOSCAR** + captions) | 28.45 | 52.35 |
> | English OF (English subsets of **mOSCAR** + English captions) | 27.95 | 53.66 |
> | OpenFlamingo-3B | 28.70 | 50.17 |
>
> These results show there is no reason to expect a significant drop of English performances when training multilingual vision-language models on mOSCAR. Similar results have been observed for text-only multilingual language models.
>
>
> [1] Awadalla et al., OpenFlamingo: An Open-Source Framework for Training Large Autoregressive Vision-Language Models (2023)
>
> [2] McKinzie et al., MM1: Methods, Analysis & Insights from Multimodal LLM Pre-training (2023)
>
> [3] Laurençon et al., OBELICS: An Open Web-Scale Filtered Dataset of Interleaved Image-Text Documents

---

> > ### Author Response · Authors · 2024-11-19
> >
> > **Q1/ Clarifying our baseline experiments**
> >
> > In Table 5, we compare the model trained on WIT + captions with the model trained on mOSCAR + captions for same amount of training data (i.e. 35M mOSCAR + 70M captions) which is an earlier checkpoint of the final multilingual Open Flamingo (trained on 50M mOSCAR + 100M captions). We modified the caption of Table 5 and we added a footnote in the paragraph starting line 441 to make this clearer (please see pdf).
> >
> > **Q2/ Performances on Multi30k**
> >
> > Our language identification filtering step mainly focuses on keeping monolingual documents as we found manually that documents with multiple languages to be very noisy (in comparison to the benefits of keeping them in the dataset). We think this is the reason why the model does not perform well on the translation task (M30K) as it may not have seen translation data during training (in contrast to WIT where some wikipedia web pages contain translations).
> >
> > **Q3/ Trade-off English-only dataset and mOSCAR**
> >
> > For the results on English-only benchmarks please see point (3) above. mOSCAR is primarily designed for multilingual training (i.e. having a model performing well in multiple languages) as such interleaved datasets only exist in English (and translating documents into target languages is a/ very expensive and b/ not very accurate at the document level). If the goal is to get state-of-the-art performance in English it should rather be trained on MINT or OBELISC which are larger in English.
> >
> >
> > **Q4/ Trade-off between enhancing few-shot capabilities and zero-shot performances**
> >
> > That is a very interesting point. We designed such an analysis by training a model on captioning data only and compared it with the model trained on mOSCAR + captioning data (see Table 4 in the paper). We found that, even in the 0-shot setting, the model trained additionally on mOSCAR was better (for similar amount of tokens seen during training), particularly on VQA benchmarks, and that the difference is less important on captioning benchmarks. This is also due to the fact that there is no good quality large-scale multilingual captioning data. In the case of English-only VLMs, [2] conducted similar experiments (see Figure 5 of their paper) by varying the ratio of mixture data (from 0% interleaved & 100% captions to the opposite). They show that a small drop on zero-shot tasks is observed when adding interleaved data (this drop linearly increases as you add more interleaved data) but you have a strong boost in few-shot learning in turn. This observation leads to different conclusions (from ours) mainly due to the fact that they trained on far more text-image pairs (captions) data in comparison to the amount of interleaved data.
> >
> > **Q5/ License**
> >
> > The dataset is released under Creative Commons CC BY 4.0 license.
> >
> > [2] McKinzie et al., MM1: Methods, Analysis & Insights from Multimodal LLM Pre-training (2023)

---

> > > ### Comment · Reviewer_Fgsw · 2024-11-27
> > >
> > > Thank the authors for the explanation and additional experiments. My concern about the English-only and non-English performance is successfully addressed. The design of the dataset is standard and not novel but I agree with the author that the analysis that shows training on interleaved multimodal data leads to better few-shot performance is valuable. I have adjusted my score accordingly.

---

### Official Review · Reviewer_Titv · 2024-11-04

**Soundness:** 4
**Presentation:** 3
**Contribution:** 4
**Rating:** 8
**Confidence:** 4

**Summary:**

This paper proposes a multimodal version of OSCAR, which is a popular multilingual dataset from CommonCrawl

## High level overview of the dataset:
- 163 languages, 303M documents, 200B tokens, 1.15B images
- This is an interleaved text-image dataset (as opposed to captioning-focused datasets)
- Trained a multilingual multimodal Flamingo-style model on the mOSCAR dataset and show it outperforms the OpenFlamingo-3B-MPT translate-test baseline across various evaluation sets.

## Overview of data collection and filtering process:
- Start from three 2023 CommonCrawl WARC files.
- Parse each document's DOM tree and use HTML tags to extract images, remove tables, etc.
- Filtering based on **size**
    - 50% removed -- too small (<500 bytes)
- Filtering based on **text** --> done on both DOM-text-node level and document-level
    - NSFW filter -- Done on document level using regex keyword matching
    - Deduplication -- Levenshtein ratio and MinHashLSH to calculate near-duplicates per language. Filter out 19\%.
    - Toxicity filter -- From list of toxic keywords in multiple languages. Filter out 1-2\%.
    - PII filter
- Filtering based on **image**
    -  URL filters -- Remove URLs that mention words like "banner", "logo", etc. Filter out 13.6 \%
    - NSFW filter -- combine multiple NSFW detectors for nudity. Filter out 0.5\%
    - CSAM filter -- Use proprietary child sexual abuse classifier. Filter out 0.07\%
    - Deduplication -- Use imagehash per language.
    - PII filter -- Detect faces
- Decontamination to remove images appearing in test sets

**Strengths:**

1. **Novelty and usefulness of contribution** -- There currently does not exist a multilingual multimodal interleaved dataset at this scale, so this artifact is something that would certainly be useful to the community.
    - This differs from other multilingual multimodal datasets (e.g. LAION-5B, mC4) since it is focused more on interleaved text+images as opposed to simple captioning.
    - This differs from other interleaved datasets (e.g. IDEFICS) because it is multilingual as opposed to pure English.
2. **Comprehensive data collection and filtering pipeline** -- The filtering process is detailed very thoroughly, and filtering is done on both text and images. Filtering was done across various features of a document, such as size, presence of keywords, image aspect ratio, etc. The pipeline is also general and scalable, so if somebody wants to do it on a larger part of CommonCrawl, it should be doable.
3. **Strong safety filters** -- This is something that is of utmost importance in designing such datasets (as we have seen from the LAION case). The paper handles this carefully, with NSFW, CSAM, and toxicity filters for both text and images. They also respect robots.txt protocols when downloading images.

**Weaknesses:**

1. **No dataset optimization** -- The paper shows that a Flamingo-style model trained with mOSCAR outperforms a translate-test baseline. This is great, but the paper didn't really mention how much iteration they did when designing this dataset. Were there multiple rounds or iterations of creating mOSCAR? There seems to be a lack of ablations, so to me it felt like the model training experiments were more of an afterthought after the dataset had already been created. (Maybe that was the intention?)
2. **No quality filter** -- This is related to the point I mentioned above. Because there is no filter specific for text quality, then the mOSCAR dataset is more of a cleaned version of CommonCrawl, as opposed to a cleaned+filtered version that is optimized for training good models.

**Questions:**

- What was the original size of the three common crawl WARC files combined? How many text tokens and images?
- I'm not very clear on the difference between the NSFW text filter and the toxicity text filter. It seems that the toxicity text filter already contains keywords for NSFW, so is this NSFW filter needed here?
- I'm a bit curious on what the main bottleneck of the pipeline is. How long would each of these steps take approximately? If someone wants to scale this up to larger datasets, which part of the process is the one that would take the most time?
---

Typos:
- L25: "train" --> "trained"
- L245 "bias" --> "biased"

Missing references:
- MINT-1T https://arxiv.org/abs/2406.11271 -- not multilingual but probably worth mentioning

**Details Of Ethics Concerns:**

I think the paper does a good job of filtering for toxicity, NSFW, nudity, etc., but adding this flag just in case.

---

> ### Author Response · Authors · 2024-11-19
>
> We would like to thank R2 for their great feedback.
>
>
> **W1/ Dataset optimization**
>
> To build mOSCAR, we mainly test different thresholds and filters in order to discard low quality content, NSFW documents & images while keeping a good tradeoff with mOSCAR diversity. We think these manual checks were a good proxy to have a good dataset for training vision language models while not training a model at multiple stages of the dataset creation. For instance, we noticed that the NSFW detector tool for images (used in previous works) contains a strong bias towards removing safe for work content such as images of women. Combining it with a detector of nudity we were able to mitigate this bias (more details in our submission line 211). We therefore did multiple iterations to build mOSCAR by using manual inspections instead of retraining a model at each step (which would have been very computationally expensive).
>
> **W2/ Quality filtering**
>
> There is no neural-based quality filter but there are many rule-based quality filters (see Section A.2 of the Supplementary Material) which were chosen manually by inspecting the data. Having a neural-based quality filter is challenging in such a multilingual setting and could be very expensive. It also comes at the expense of the diversity of the dataset.
>
>
>
> **Questions:**
>
> **Q1: Original size of the 3 Common Crawl dumps**
>
> We estimate that there are approximately 20 trillion tokens and 60 billion images in the three CommonCrawl dumps we processed.
>
> **Q2: Difference NSFW and Toxicity filters**
>
> Toxicity word lists include some NSFW words. However, as we want to get a dataset as safe as possible, we additionally apply a NSFW regular expression (targeting English content) to further remove NSFW content (that may not have been removed with the toxicity filter only).
>
> **Q3: Main bottleneck of the pipeline**
>
> Definitely the Web Archive Content processing pipeline (i.e. first step, going through all CommonCrawl files) and downloading the images. These steps are super slow, require good hardware and network bandwidth. For instance, collecting images for 3 CommonCrawl dumps took approximately 2 months on our setup.
> Thank you for pointing out the typos and the missing reference that we will add in the related work (in the “English image-text datasets” paragraph).

---

### Official Review · Reviewer_QB5F · 2024-11-12

**Soundness:** 3
**Presentation:** 3
**Contribution:** 3
**Rating:** 3
**Confidence:** 4

**Summary:**

The paper presents the creation and utilization of mOSCAR, a novel large-scale multilingual and multimodal dataset for training Multimodal Large Language Models (mLLMs). The dataset, which includes 303 million documents with 1.15 billion images spanning 163 languages, was created to address the limitations of existing datasets that are often monolingual or only moderately multilingual. By applying a series of filtering steps to enhance safety, diversity, and quality, mOSCAR aims to support multilingual, in-context learning capabilities. The authors demonstrate mOSCAR's benefits by training models that outperform baseline models across multiple multilingual, multimodal benchmarks. The proposed dataset seeks to address the scarcity of high-quality, large-scale multilingual datasets by employing rigorous data collection, filtering, and decontamination techniques. The authors validate mOSCAR's efficacy by training and evaluating two models: one on mOSCAR and captioning data, and another solely on captioning data. The mOSCAR-trained model shows marked improvements, particularly in few-shot learning across multilingual tasks.

**Strengths:**

mOSCAR is derived from Common Crawl and utilizes sophisticated methods to filter and refine data. Steps include document deduplication, toxicity filtering, PII removal, and NSFW detection, ensuring a robust dataset suitable for mLLMs. Covering 163 languages, mOSCAR significantly expands the multilingual training dataset landscape, making it inclusive of languages that are often underrepresented in large-scale datasets. The authors conducted extensive benchmark tests, demonstrating that models trained with mOSCAR achieve superior performance in few-shot learning and across a variety of image-text tasks compared to models trained solely on captioning data. The authors plan to make the dataset public, promoting accessibility and encouraging further research in multilingual and multimodal model training. With meticulous filtering to maintain both quality and diversity, the paper ensures that mOSCAR addresses common issues like NSFW content, toxic language, and duplication, which enhances its usability in ethical AI research.

**Weaknesses:**

1. Limited manual validation (1,000 images for NSFW, 100 documents for toxic content)
2. English-centric regex patterns may miss unsafe content in other languages
3. Binary toxicity detection requiring two distinct toxic words could miss subtle harmful content
4. Document-level NSFW filtering is overly aggressive, potentially discarding valuable safe content
5. Reliance on regex and wordlists could be improved with neural-based approaches
6. Fixed character (300) and node count (5) thresholds may not accommodate different writing systems
7. Language identification may underperform on code-switched or multilingual content
8. Limited evaluation of data quality and distribution across the 163 languages, particularly for low-resource languages
9. No systematic evaluation of text-image alignment quality
10. Lack of content relevance metrics within documents

**Questions:**

As weakness

---

> ### Author Response · Authors · 2024-11-19
>
> We would like to thank R3 for their review.
>
>
> **W1/ Limited manual evaluation**
>
> We did our best effort to inspect examples manually. As we randomly sampled from the dataset for the manual analysis, looking at 1,000 examples should be enough to approximate well the whole dataset. Furthermore, since its creation, our dataset has been used by multiple people and we have not received any reports regarding the presence of NSFW content or toxic content yet. That said, we additionally annotated 900 more documents for toxic content to have a more accurate estimation of the content removed and we found a total of 568 toxic documents among the 1,000 removed ones (which confirms the previous estimation).
>
> **W2/ English centric patterns to remove unsafe content**
>
> For other languages (including English) we conduct some toxic content removal filtering steps using a dictionary of possible toxic words (including NSFW words) (see Section 3.3, line 178, in the paper for more details).
>
> **W3/ Toxic content filter could miss subtle harmful content**
>
> We agree that we could have missed some harmful content. We did our best to remove as much harmful content as possible while maintaining high dataset quality and diversity. We used a state-of-the-art method to detect harmful content in multilingual text data (the NLLB method described in Section 3.3, line 178). Moreover, if we had removed all documents containing at least one word from the list (instead of two different words as we did), we would have introduced huge biases into the dataset. For example, all documents containing the words “jews” or “breast” would have been removed from mOSCAR as “jews” is sometimes found in hateful content on the internet and “breast” can be used in a sexual meaning.
>
> **W4/ Document-level NSFW filtering is overly aggressive**
>
> As mentioned above, it is a matter of tradeoff between recall and precision, NSFW is very sensitive and we did our best to remove it from mOSCAR by favouring recall over precision. It is indeed not possible to remove as much NSFW content as possible while keeping all safe content. We chose to do so for safety reasons and also because some datasets (e.g. LAION5B) were recently found to have lots of NSFW content (including child sexual abuse material) in it.
>
> **W5/ Reliance on regex and words lists can be improved with neural-based approaches**
>
> Relying on neural based approaches is very useful if it is light, affordable and offers better results than regex rules. In our experiments, whenever we use regex and word lists, either there were no existing neural based approaches or they did not provide sufficiently better results (if at all) in comparison to their cost of compute.
>
> **W6/ Fixed characters and node count may not be appropriate for all writing systems**
>
> As mOSCAR is designed as a document-level dataset, we found that documents smaller to 300 characters were almost always not relevant or too short (same for 5 nodes). We also found empirically that 300 characters was a good tradeoff to remove irrelevant documents in most languages.
>
> **W7/ Language identification for code-switched content**
>
> We chose to focus our language identification method on monolingual documents as we found empirically that keeping documents with code-switched content implies keeping lots of low quality documents.
>
> **W8/ Limited evaluation of the quality for all languages**
>
> It is quite challenging to perform quality evaluation for all languages except by conducting human evaluation, which is very expensive in such a setting. This is why we focused on English evaluation and compared against well-known English-only document-level datasets.
>
> **W9/ No systematic evaluation for image-text alignment**
>
> An image-text alignment quality analysis can be found in Section A.4 (see Supplementary Material). We found that the images in the document are generally related to surrounding text in mOSCAR, according to NLLB-CLIP similarity, which means that text and images close to each other in a document are well aligned within the document.
>
> **W10/ Lack of content relevance metrics within documents**
>
> We provide a quantitative analysis of the semantic content of the English documents in Section 4.1.1. We found that mOSCAR is on average more semantically diverse than other comparable datasets.

---

### Official Review · Reviewer_DHGk · 2024-11-12

**Soundness:** 3
**Presentation:** 3
**Contribution:** 3
**Rating:** 6
**Confidence:** 3

**Summary:**

This paper yields a powerful Multimodal Multilingual pretraining dataset for MLLM's. They choose a standard set of filters and approaches to arrive a higher quality. Downstream models trained on mOSCAR are empirically stronger than existing models.

**Strengths:**

1. The choice of filters was discussed thoroughly. Relative to other literature in the field, it is clear that an exhaustive search over the space of possible data filtering methodologies was done.
2. The empirical results are quite promising. The set of evaluations/benchmarks is extensive and exhaustive. Moreover, the performance across different shotting of the models demonstrates significant improvmenets over existing datasets.
3. I found the discussion around diversity measurements and comparison to existing datasets to be very well done. The discussion how multilinguality vs English-only diversity needs to be accounted for was also enlightening.
4. The prevention against data poisoning is appreciated and important.

**Weaknesses:**

1. I found some of the choices, while fairly standard, for the design of this framework to be lacking in terms of thought in terms of their design. While most of the decisions are cited, several decisions could have been had more analysis.
2. Ablations on the filters would have been very interesting to see. This includes the joint image-text filtering.

**Questions:**

N/A

---

> ### Author Response · Authors · 2024-11-19
>
> We would like to thank R4 for their review.
>
> **W 1&2/ Analysis and ablation study of the dataset construction**
>
> Our choices are based on the literature of multilingual datasets and multimodal datasets. We additionally manually check the content removed and the content remaining for all our filtering steps to be sure we maintain a good tradeoff between quality, diversity and safety. We were unfortunately not able to conduct an ablation study for all our choices as this would mean training a new model at different filtering steps, which would require a huge compute budget that we do not have access to. Regarding our text-image filtering step, as we remove lots of irrelevant text and image nodes within documents while keeping relevant ones, we are fairly confident that we improve the overall quality of mOSCAR. An analysis of the quality of text-image alignment can be found in Section A.4 in the Supplementary Material where we show that images close to text nodes within a document are generally highly similar.

---

### Author Response · Authors · 2024-11-25

We would like to thank all reviewers for their valuable feedback and hope that we have clarified their concerns. As the discussion period comes to an end, we would appreciate if the reviewers could come back to their reviews and acknowledge our rebuttal.

---

### Meta-Review · Area_Chair_GhSD · 2024-12-23

**Metareview:**

This paper presents mOSCAR, described as the first large-scale multilingual and multimodal web-crawled document corpus. The key claims are that it covers 163 languages with 303M documents, 200B tokens and 1.15B images, and that models trained on mOSCAR show improved few-shot learning capabilities. While the paper addresses an important need for multilingual multimodal training data, several critical weaknesses make it unsuitable for acceptance at this time. The filtering methodology raises serious concerns, particularly for non-English content where safety filters may miss harmful content due to English-centric patterns. The manual validation is extremely limited (only 1,000 images for NSFW, 100 documents for toxic content initially), making it difficult to assess dataset quality and safety. The binary toxicity detection requiring two distinct toxic words could miss subtle harmful content, while document-level NSFW filtering may be overly aggressive in discarding potentially valuable safe content. The experimental evaluation also has significant gaps - there's limited analysis of text-image alignment quality, no systematic evaluation of content quality across the 163 languages (especially low-resource ones), and unclear baseline comparisons. While the authors attempted to address some concerns during rebuttal by adding 900 more document evaluations and new experimental results, these additions still leave major questions about dataset safety and quality unanswered. The review scores (3, 6, 6, 8) reflect these serious concerns. Thus I vote to reject

**Additional Comments On Reviewer Discussion:**

The reviewers raised several significant concerns during rebuttal. Reviewer QB5F highlighted multiple critical issues around safety filtering, language coverage, and quality assessment - while the authors added some manual evaluations, their responses did not fully address the fundamental concerns about English-centric filtering and limited quality validation. Reviewer Titv questioned dataset optimization and quality filtering - the authors acknowledged not using neural-based quality filters due to computational constraints. Reviewer Fgsw requested clarification on English performance impact - while the authors provided new experimental results showing comparable English performance, this addresses only one of many concerns. Most critically, the authors' responses around safety filtering revealed concerning limitations: relying primarily on regex patterns and word lists rather than more sophisticated approaches, using fixed character thresholds that may not work across writing systems, and having very limited manual validation. The additional 900 document evaluations, while welcome, represent a tiny fraction of the 303M total documents. While some reviewers were satisfied with certain clarifications, the core issues around dataset safety and quality validation remain insufficiently addressed

---

### Decision · Program_Chairs · 2025-01-22

Reject